# V-OCBF: Learning Safety Filters from Offline Data via Value-Guided Offline Control Barrier Functions

**Mumuksh Tayal**[*]                                                           *mumukshtayal@iisc.ac.in*
*Center for Cyber Physical Systems (CPS)*
*Indian Institute of Science (IISc) Bengaluru, India*

**Manan Tayal**[*]                                                           *t-manantayal@microsoft.com*
*Microsoft Research, India*

**Aditya Singh**                                                           *singhadi@seas.upenn.edu*
*Department of Electrical and Systems Engineering*
*University of Pennsylvania, Philadelphia, USA*

**Shishir Kolathaya**                                                           *shishirk@iisc.ac.in*
*Center for Cyber Physical Systems (CPS)*
*Indian Institute of Science (IISc) Bengaluru, India*

**Ravi Prakash**                                                           *ravipr@iisc.ac.in*
*Center for Cyber Physical Systems (CPS)*
*Indian Institute of Science (IISc) Bengaluru, India*

[*] **denotes equal first author contribution**

**Reviewed on OpenReview:** `https://openreview.net/forum?id=PGO9mpIyyb`

## Abstract

Ensuring safety in autonomous systems requires controllers that aim to satisfy state-wise constraints without relying on online interaction.While existing Safe Offline RL methods typically enforce soft expected-cost constraints, they struggle to ensure strict state-wise safety. Conversely, Control Barrier Functions (CBFs) offer a principled mechanism to enforce forward invariance, but often rely on expert-designed barrier functions or knowledge of the system dynamics. We introduce Value-Guided Offline Control Barrier Functions (V-OCBF), a framework that learns a neural CBF entirely from offline demonstrations. Unlike prior approaches, V-OCBF does not assume access to the dynamics model; instead, it derives a recursive finite-difference barrier update, enabling model-free learning of a barrier that propagates safety information over time. Moreover, V-OCBF incorporates an expectile-based objective that avoids querying the barrier on out-of-distribution actions and restricts updates to the dataset-supported action set. The learned barrier is then used with a Quadratic Program (QP) formulation to synthesize real-time safe control. Across multiple case studies, V-OCBF yields substantially fewer safety violations than baseline methods while maintaining strong task performance, highlighting its scalability for offline synthesis of safety-critical controllers without online interaction or hand-engineered barriers.

## 1 Introduction

Ensuring the safety of autonomous systems is essential for their reliable and widespread deployment. From household service robots to autonomous vehicles and aerial drones, these systems increasingly operate in complex and unstructured environments where unsafe behavior can lead to irreversible consequences. As

autonomy becomes deeply integrated into transportation, manufacturing, and healthcare, guaranteeing that such systems operate within well-defined safety boundaries is critical for reliability, and long-term adoption.

Reinforcement learning (RL) has emerged as a powerful paradigm for enabling autonomous systems to acquire sophisticated control behaviors. However, in safety-critical domains, naïve RL exploration can be hazardous. Although constrained RL (CRL) (Achiam et al., 2017; Altman, 2021; Alshiekh et al., 2018; Zhao et al., 2023) methods attempt to incorporate safety constraints during learning, they typically require extensive online interaction with the environment. However, most previous studies focus on online RL setting (Liu et al., 2024), which suffers from serious safety issues in both training and deployment phases, especially for scenarios that lack high-fidelity simulators and require real system interaction for policy learning. As a result, there is a growing interest in synthesizing safe policies using offline RL or imitation learning (Levine et al., 2020; Kumar et al., 2020). Nevertheless, most online and offline safe RL approaches (Xu et al., 2022; Ciftci et al., 2024; Stooke et al., 2020) treat safety as a *soft constraint* and regulate only the expected cumulative constraint violations. Such probabilistic constraints are insufficient for applications that demand strict state-wise safety, where even a single violation is unacceptable. Furthermore, jointly optimizing performance and safety from static datasets often leads to unstable training dynamics and overly conservative behavior, particularly when safety-critical transitions are sparsely represented(Lee et al., 2022).

Control-theoretic tools provide an alternative and more rigorous foundation for safety. In particular, Control Barrier Functions (CBFs) (Ames et al., 2014) offer a principled mechanism to enforce instantaneous safety constraints by guaranteeing the forward invariance of a prescribed safe set. When combined with learning-based controllers, CBFs serve as minimally invasive safety filters that adjust nominal actions only when necessary to prevent constraint violations. Their integration with Quadratic Program (QP) based controllers enables real-time implementation with modern optimization solvers. Consequently, CBF-based controllers have been successfully applied to a wide range of safety-critical tasks, including adaptive cruise control (Ames et al., 2014), aerial robotics (Wu & Sreenath, 2016; Tayal et al., 2024a), and legged locomotion (Nguyen & Sreenath, 2015). In all of these applications, the performance and safety guarantees fundamentally depend on the quality of the underlying CBF.

Constructing valid CBFs, however, is a challenging problem. Hand-crafting barrier functions requires deep system knowledge and does not scale well to high-dimensional or partially known dynamical systems. This has motivated significant interest in Neural Control Barrier Functions (NCBFs), which leverage the expressive power of neural networks to approximate complex safe sets. A variety of techniques have been proposed for learning NCBFs, including SMT-based synthesis (Abate et al., 2021; 2020), mixed-integer programming (Zhao et al., 2022), nonlinear optimization (Zhang et al., 2023), and loss-based training methods (Dawson et al., 2022; 2023; Tayal et al., 2024b; 2025). Other recent approaches learn CBFs from value functions associated with nominal policies (So et al., 2024). However, most of these methods rely on online interaction to collect informative samples or refine the barrier, which is often infeasible in safety-critical settings.

Recent work has explored learning Control Barrier Functions (CBFs) from offline demonstrations (Robey et al., 2020; Castañeda et al., 2023; Tabbara & Sibai, 2025). Existing methods either fit CBFs only on expert trajectories or rely on data-likelihood measures to filter unsafe samples, which limits their ability to generalize beyond the demonstrated states. Uncertainty-aware approaches address distributional mismatch but often become overly conservative. Overall, current offline CBF learning methods are closely tied to the empirical data distribution and do not explicitly reason about future system evolution, resulting in conservative safety.

This paper introduces Value-Guided Offline Control Barrier Functions (V-OCBF), a novel framework designed to overcome key limitations of existing offline RL and CBF-based approaches. We derive a model-free finite-difference barrier recursion and establish that, in the idealized setting without learning or approximation errors, enforcing this update provides a formal one-step forward-invariance safety guarantee for control-affine systems. In addition, we propose an expectile-based learning objective that allows the synthesized safe policy to improve over the behavior policy in the dataset while never querying the barrier on out-of-distribution actions, ensuring stable and reliable offline learning. To summarise, the main contributions of this work are as follows:

1. We propose V-OCBF, a novel framework for learning safe controllers entirely from offline demonstrations.

2. We derive a model-free finite-difference barrier recursion and prove that adherence to this update guarantees one-step forward invariance for any control-affine system.

3. We introduce an expectile-based objective that improves upon the behavior policy without evaluating the barrier outside the dataset action support.

4. Across diverse systems, including high-dimensional Safety Gymnasium (Ji et al., 2023) tasks, V-OCBF consistently outperforms constrained offline RL and neural CBF baselines in both safety and reward.

## 2 Related Works

### 2.1 Safe Offline RL

Safe RL has traditionally been studied in the online setting, where most methods rely on Lagrangian-based constrained optimization (Chow et al., 2017; Tessler et al., 2018; Stooke et al., 2020). These approaches regulate safety through cost penalties, treating constraints as soft and therefore allowing non-zero violation risk. CPO (Achiam et al., 2017) provides theoretical guarantees during updates through a trust-region mechanism, but still cannot ensure strict safety throughout online learning. These limitations motivate shifting toward safe offline RL , where policies are synthesized from static data to avoid unsafe exploration. Among early offline-safe RL methods, CPQ (Xu et al., 2022) assigns large costs to unsafe or out-of-distribution actions, but this can distort the value function and reduce generalization (Li et al., 2023). COptiDICE (Lee et al., 2022), building on DICE objectives (Lee et al., 2021), inherits the difficulties of residual-gradient learning (Baird, 1995). More recent work integrates safety considerations into Decision Transformer (Chen et al., 2021) or Diffuser models (Janner et al., 2022), enabling sequence-modeling-based safety (Liu et al., 2023; Lin et al., 2023; Zheng et al., 2024). However, these architectures are computationally expensive and challenging to scale. Overall, while offline RL mitigates unsafe online interaction, existing methods primarily enforce soft constraints and often struggle when unsafe transitions are underrepresented or missing from the dataset.

### 2.2 Offline Neural CBFs

Complementary to RL-based strategies, several recent methods aim to learn Control Barrier Functions (CBFs) directly from offline demonstrations. The approach in (Robey et al., 2020) enforces differentiable barrier conditions on expert trajectories but does not account for out-of-distribution (OOD) states, limiting generalization beyond demonstrated data. The iDBF framework (Castañeda et al., 2023) mitigates unsafe generalization by filtering low-likelihood states and actions using a Behavior Cloned policy, although this reliance on BC likelihood restricts coverage when demonstrations do not span the full safe set. Conservative CBFs (Tabbara & Sibai, 2025) incorporate uncertainty-aware penalties to avoid optimistic predictions on OOD states, but this often yields overly cautious certificates that substantially underestimate the true safe region. While these approaches represent important progress, they fundamentally rely on the empirical demonstration distribution and do not explicitly model how system dynamics may evolve toward unsafe states. Consequently, they tend to produce conservative CBFs that under-approximate the true safe set.

### 2.3 Positioning of Our Work

V-OCBF positions itself primarily as a framework for **learning safe control policies from offline datasets**, utilizing Control Barrier Functions (CBFs) as a learned intermediate representation to enforce strictly safe behavior. Unlike safe offline RL methods (Lee et al., 2022; Liu et al., 2023; Lin et al., 2023; Zheng et al., 2024), which rely on soft constraints, we directly construct a safety certificate and a safe policy that guarantees one-step forward invariance, in the idealised setting. In contrast to model-based safe offline approaches that rely on learned dynamics both for training neural CBFs and during inference (Robey et al.,

2020; Castañeda et al., 2023; Tabbara & Sibai, 2025), V-OCBF employs a model-free value recursion that models future safety evolution and uses dynamics only at the time of inference for local Lie derivative computation in the CBF-QP, rather than for policy optimization or imagined rollouts. This design prevents the policy from extrapolating into action regions unsupported by the dataset and preserves the OOD robustness of our expectile-based objective. Overall, our method combines the distributional robustness of offline RL with the rigor of CBF-based control to obtain an offline-learned, actuation-aware neural CBF without requiring online rollouts.

## 3 Background and Problem Setup

We consider a control-affine nonlinear dynamical system defined by the state $x(t) \in \mathcal{X} \subseteq \mathbb{R}^n$, the control input $u(t) \in \mathbb{U} \subseteq \mathbb{R}^m$, and governed by the following dynamics:

$$\dot{x}(t) = f(x(t)) + g(x(t))u(t), \tag{1}$$

where $f : \mathbb{R}^n \to \mathbb{R}^n$ and $g : \mathbb{R}^n \to \mathbb{R}^{n \times m}$ are locally Lipschitz continuous functions. We are given a set $\mathcal{C} \subseteq \mathcal{X}$ that represents the *safe states* for the system and a failure set $\mathcal{F} \subseteq \mathcal{X}$ that represents the set of unsafe states for the system (e.g., obstacles for an autonomous ground robot). Furthermore, the system is controlled by a Lipschitz continuous control policy $\pi : \mathbb{R}^n \to \mathbb{R}^m$. Our focus lies in ensuring the safety of this dynamical system, which is formally defined as follows:

**Definition 1** (Safety). *A dynamical system is considered safe if the set, $\mathcal{C} \subseteq \mathcal{X} \subseteq \mathbb{R}^n$, is positively invariant under the control policy, $\pi$, i.e, $x(0) \in \mathcal{C}, u(t) = \pi(x(t)) \implies x(t) \in \mathcal{C}, \forall t \geq 0$.*

Since, $\mathcal{F} \subseteq \mathcal{X} \setminus \mathcal{C}$, it can be trivially shown that $x(t) \in \mathcal{C} \implies x(t) \notin \mathcal{F} \ \forall \ t \geq 0$. Using this premise, we define the main objective of this paper:

**Objective 1.** *Our objective is to synthesize a safe policy $\pi_{\text{safe}} : [t, T] \times \mathcal{X} \to \mathbb{U}$ such that the resulting closed-loop system satisfies the positive invariance property specified in Definition 1.*

### 3.1 Control Barrier Functions

Control Barrier Functions (Ames et al., 2014; 2017) are widely used to synthesize control policies with positive invariance guarantees, thereby ensuring system safety. The initial step in constructing a Control Barrier Function (CBF) involves defining a continuously differentiable function $B : \mathcal{X} \to \mathbb{R}$, where the *super-level set* of $B$ corresponds to the safe region $\mathcal{C}$. This leads to the following representation:

$$\mathcal{C} = \{x \in \mathcal{X} : B(x) \geq 0\}, \quad \mathcal{X} \setminus \mathcal{C} = \{x \in \mathcal{X} : B(x) < 0\}. \tag{2}$$

The interior and boundary of $\mathcal{C}$ are further specified as:

$$\text{Int}(\mathcal{C}) = \{x \in \mathcal{X} : B(x) > 0\}, \quad \partial\mathcal{C} = \{x \in \mathcal{X} : B(x) = 0\}. \tag{3}$$

The function $h$ qualifies as a valid Control Barrier Function if it satisfies the following definition:

**Definition 2** ((Ames et al., 2017)). *Given a control-affine system $\dot{x} = f(x) + g(x)u$, the set $\mathcal{C}$ defined by equation 2, with $\frac{\partial B}{\partial x}(x) \neq 0$ for all $x \in \partial\mathcal{C}$, the function $B$ is called the Control Barrier Function (CBF) defined on the set $\mathcal{X}$, if there exists an extended class-$\mathcal{K}$ function $\kappa$ such that for all $x \in \mathcal{X}$:*

$$\max_{u \in \mathbb{U}} \left[ \underbrace{\mathcal{L}_f B(x) + \mathcal{L}_g B(x)u}_{\dot{B}(x,u)} + \kappa\left(B(x)\right) \right] \geq 0, \tag{4}$$

*where $\mathcal{L}_f B(x) = \frac{\partial B}{\partial x} f(x)$ and $\mathcal{L}_g B(x) = \frac{\partial B}{\partial x} g(x)$ are the Lie derivatives and $n$ is the dimension of the system.*

As established in (Ames et al., 2017), any Lipschitz continuous control law $\pi(x)$ that satisfies the condition $\dot{B} + \kappa(B) \geq 0$ guarantees the system's safety when $x(0) \in \mathcal{C}$. Additionally, if the initial state $x(0)$ lies outside $\mathcal{C}$, this condition ensures asymptotic convergence to the safe set $\mathcal{C}$.

While CBFs provide a principled framework to guarantee safety, their practical deployment is hindered by the lack of general methods for constructing valid barrier functions. As a result, practitioners typically resort to handcrafted or domain-specific CBFs, which can yield overly conservative safe sets. Furthermore, in the presence of control bounds, a nominal CBF may conflict with feasibility requirements, causing the corresponding CBF-QP to become infeasible.

### 3.2 Control Barrier Value Function (CBVF)

To overcome the limitations inherent in classical CBF formulations, (Choi et al., 2021) proposed the *Control Barrier Value Function* (CBVF), which integrates Control Barrier Functions with Hamilton–Jacobi (HJ) Reachability (Bansal et al., 2017). We begin by encoding the safety specification using a Lipschitz continuous function $\ell : \mathbb{R}^n \to \mathbb{R}$, where the failure set is defined as $\mathcal{F} := \{x \in \mathcal{X} \mid \ell(x) < 0\}$. Under this construction, a CBVF $B : \mathcal{X} \to \mathbb{R}$ is defined as the viscosity solution of the following Hamilton–Jacobi–Bellman Variational Inequality (HJB-VI):

$$\min\left\{ \max_{u \in \mathbb{U}}(\nabla B(x) \cdot (f(x) + g(x)u)), \; \ell(x) - B(x) \right\} = 0, \tag{5}$$

with boundary condition $B(x)|_{t=0} = \ell(x)|_{t=0}$. The resulting value function induces a forward-invariant safe set $\mathcal{C} := \{x \in \mathcal{X} \mid B(x) \geq 0\}$ and ensures that admissible controls $u \in \mathbb{U}$ satisfy the following Lie-derivative condition:

$$\max_{u \in \mathbb{U}} \left[ L_f B(x) + L_g B(x)u + \kappa(B(x)) \right] \geq 0. \tag{6}$$

**Safe Controller Synthesis using CBVFs:** Quite often, we have a reference control policy, $\pi_{ref}(x)$, designed to meet the performance requirements of the system. However, such controllers often lack safety guarantees. To ensure the system meets its safety requirements while preserving performance, the reference controller must be minimally adjusted to incorporate safety constraints. This adjustment can be accomplished using the Control Barrier Value Function-based Quadratic Program (CBVF-QP), described as follows:

$$\pi_{\text{safe}}(x) = \min_{u \in \mathbb{U} \subseteq \mathbb{R}^m} \|u - \pi_{ref}\|^2$$
$$\text{s.t. } \mathcal{L}_f B(x) + \mathcal{L}_g B(x)u + \kappa\left(B(x)\right) \geq 0. \tag{7}$$

The CBVF-QP framework facilitates the synthesis of a provably safe control policy, $\pi_{\text{safe}}(x)$, while staying close to the reference controller to preserve system performance.

**Challenges in CBVF Synthesis:** Traditional approaches compute Control Barrier Value Functions using grid-based HJ reachability methods (Mitchell, 2005), which are fundamentally limited by the curse of dimensionality. Recent efforts have attempted to overcome these issues by learning CBVFs through online reinforcement learning (So et al., 2024); however, as discussed in Section 1, such approaches require extensive online interaction, rendering them unsuitable for safety-critical systems. Furthermore, existing neural CBF methodologies (Abate et al., 2020; Zhang et al., 2023; Tayal et al., 2024b) typically assume access to accurate system dynamics, an assumption that does not hold for many real-world platforms. These limitations motivate a shift toward using offline demonstrations, either sourced from public datasets (Liu et al., 2024; Sun et al., 2020) or collected in controlled settings where safety can be guaranteed. Building on this premise, we refine Objective 1 as follows:

**Objective 2.** *Our objective is to synthesize a Control Barrier Value Function $B : \mathcal{X} \to \mathbb{R}$ directly from an offline dataset of demonstrations $\mathcal{D}$, such that the resulting CBVF-QP controller $\pi_{\text{safe}}$ in equation 7 aims to satisfy the positive invariance property specified in Definition 1.*

## 4 Methodology

Having introduced the CBVF formulation in the previous section, we now describe a practical methodology for synthesizing a valid Control Barrier Value Function and thereby the safe controller from offline demonstrations $\mathcal{D}$. Our objective is to construct a data-driven approximation of the viscosity solution of the HJB-VI equation 5 without relying on known system dynamics or online interaction. The key idea is to re-interpret it through a finite-difference barrier recursion compatible with demonstration data, and to use this recursion to learn a value-guided barrier function that inherits forward invariance.

### 4.1 Finite-Difference Barrier Synthesis

To approximate the CBVF using trajectories in $\mathcal{D}$, we consider a finite difference recursive version of equation equation 5. Given a trajectory $\{x_t, u_t\}_{t=0}^{N}$, we define the finite-difference barrier update

$$B(x_t) = \min\left\{\ell(x_t), \max_{u_t} B(x_{t+1})\right\}, \qquad \forall t \in \{0, 1, 2, \dots\}, \tag{8}$$

where, $x_t = x(t)$, $u_t = u(t)$ and $x_{t+1} = x(t + \Delta t)$. This recursion has two significant advantages. First, it enables us to learn a barrier directly from data without requiring $f$ and $g$. Second, under mild regularity assumptions, equation 8 preserves the forward invariance property of the CBVF. The derivation of the recursive equation is provided in Appendix A.1. Intuitively, the recursion encodes the principle that a state is safe if and only if its immediate successor is safe or it lies outside the unsafe set as specified by $\ell(x)$.

To represent this barrier function, we parameterize a neural network $B_\psi^{\mathcal{D}}(x)$ with $\psi$ utilizing the universal approximation property. However, directly solving for the recursion equation 8 can lead to degenerate solutions. For instance, setting $B_\psi^{\mathcal{D}}(x) = c$ for a sufficiently small constant $c$ satisfies equation 8 but clearly does not satisfy the CBVF conditions in equation 5. This pathology is analogous to the non-contractive behavior of undiscounted value iteration in MDPs. Following the approach in (Fisac et al., 2019), we incorporate a discounted finite-difference value to avoid such trivial solutions:

$$B_{Target}^{\mathcal{D}} = (1 - \gamma)\,\ell(x) + \gamma \min\left\{\ell(x), \max_u B_\psi^{\mathcal{D}}(x')\right\}, \tag{9}$$

where $x = x_t$, $x' = x_{t+1}$ and $u = u_t$ and $\gamma \to 1$. This discounted recursion ensures contraction, promotes stable learning, and prevents the network from collapsing to uniformly unsafe or uniformly safe solutions. To avoid evaluating barrier targets at out-of-distribution actions, we remove the maximization over actions from the equation 9 and use only the demonstrated action in each transition. While this prevents unsupported queries, the resulting estimate reflects the safety profile of the behaviour policy that generated the data. Consequently, this produces a behaviour-induced barrier, which is typically sub-optimal because it ignores other admissible actions that could yield larger safe-set estimates.

### 4.2 Avoiding Out-of-Distribution Actions in Offline Learning

The naïve regression objective in equation 9 with the maximization over actions dropped, fits $B_\psi^{\mathcal{D}}$ to the mean of the demonstrated next-state targets, but this corresponds to the behavior-induced barrier and yields overly conservative safe sets. Ideally, if we assume unlimited capacity and no sampling error, the optimal parameters should satisfy, $B(x) \approx \mathbb{E}_u\left[\min\left\{\ell(x), \max_u B(x')\right\}\right]$. However, such unconstrained maximization can result in actions that are never observed in the dataset.

Since offline data only provides information about those actions selected by the behavior policy, evaluating values (or barrier targets) using unsupported actions can distort learning because the corresponding transitions are not grounded in the dataset. Subsequently, motivated by the insights of Implicit Q-Learning (IQL) (Kostrikov et al., 2022), under idealized conditions, deterministic dynamics and disturbance-free data, prioritizing the best observed next-state outcomes can yield strong empirical performance, since emphasizing high-value targets leads to safer barrier estimates. However, this naïve maximization is brittle in practice: rare poor actions or stochastic disturbances can occasionally produce optimistic next states, and focusing

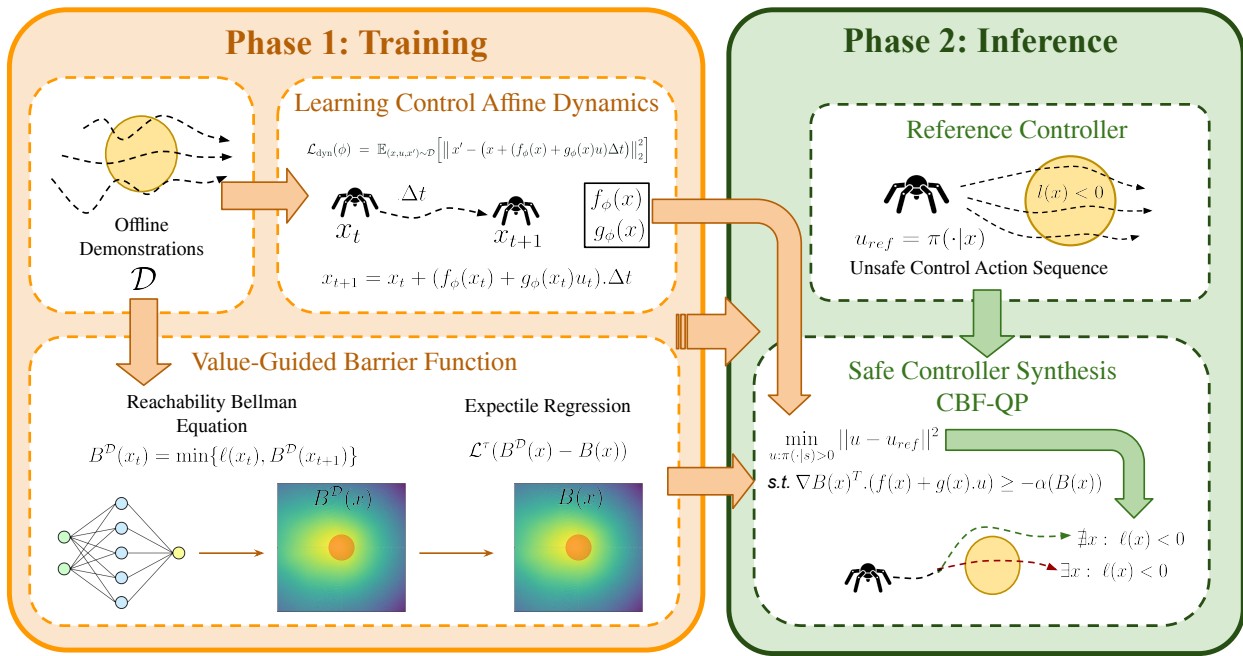

Figure 1: **Framework Overview:** *(Left)*: We learn value guided barrier function with reachability based bellman equation for learning optimal safe region and apply expectile regression for OOD case handling, when learning from Offline Dataset. *(Right)*: Inferencing with CBF-QP using learned barrier function as valid CBF to rollout safe step-wise actions, with any reference controller.

only on the single best outcome can cause the learner to overfit such spurious transitions. Expectile regression provides a simple remedy by emphasizing high next-state barrier values that actually appear under dataset-supported actions, rather than committing to a single extreme outcome. In this way, it preserves the benefit of prioritizing good actions while remaining robust to noise and avoiding evaluations on unseen $(x, u)$ pairs. For details and sensitivity to $\tau$, see Appendix A.2 and Appendix C.3, respectively. This allows us to perform a principled value-style backup over the dataset control action without extrapolating to unsafe or unobserved actions. This IQL-inspired method shows that expectile regression produces a value function that reflects the values induced by the behavior policy without requiring an explicit behavior model. This prevents the learning target from being influenced by actions that lie outside the dataset support, while still capturing the highest feasible values supported by the demonstrations.

Following this principle, we estimate a CBVF, $B_\theta$, with $\theta$ as the Neural Network parameters, that reflects the safety values implied by demonstrated actions. Formally, we minimize the expectile loss

$$\mathcal{L}_B(\theta) \;=\; \mathbb{E}_{(x,u,x')\sim\mathcal{D}}\big[\mathcal{L}^\tau\big(B^{\mathcal{D}}_{Target}(x) - B_\theta(x)\big)\big], \tag{10}$$

where $\mathcal{L}^\tau(y) = |\tau - \mathbf{1}(y < 0)|\,y^2$ is the $\tau$-expectile loss used in (Kostrikov et al., 2022) and we replace the original objective term $B^{\mathcal{D}}_{Target}$ from equation 9 with the new objective $B^{\mathcal{D}}_{Target} = (1-\gamma)\ell(x) + \gamma \min\{\ell(x), B_\theta(x')\}$ which incorporates $B_\theta(x')$ in place of $\max_u B^{\mathcal{D}}_\psi(x')$. Intuitively, a higher expectile level $\tau$ places greater weight on underestimation errors than overestimation errors, pushing $B_\theta$ toward the upper envelope of safety values supported by the dataset. Thus, $\tau$ controls how aggressively the learned barrier emphasizes high, data-supported safety values without extrapolating to unseen actions. In practice, this is implemented by computing the expectile loss over the empirical state-action transitions within each sampled minibatch, thereby restricting the updates to the dataset-supported action set without requiring explicit support construction. The barrier function thus obtained, $B_\theta$, is our proposed *Value-guided Offline Control Barrier Function* (V-OCBF). The complete learning procedure for $B^{\mathcal{D}}_\psi$ and the V-OCBF $B_\theta$ is summarized in Algorithm 1 and can be visually seen in Fig. 1.

---

**Algorithm 1** Learning V-OCBF from Offline Demonstrations

---

**Require:** Offline dataset $\mathcal{D}$, expectile level $\tau$, batch size $m$, learning rates $\eta_\phi, \eta_\theta$, discount factor $\gamma$
1: Initialize $\psi, \theta$ and optimizers
2: **for** epoch $= 1$ **to** $N$ **do**
3:      **for** minibatch $\mathcal{B} \subset \mathcal{D}$ **do**
4:          $\mathcal{L}_B \leftarrow \frac{1}{|\mathcal{B}|} \sum_{(x,u,x') \in \mathcal{B}} \mathcal{L}^\tau \left( B_{Target}^{\mathcal{D}}(x) - B_\theta(x) \right)$             $\triangleright$ From Eq. 10
5:          $\theta \leftarrow \text{AdamStep}(\theta, \nabla_\theta \mathcal{L}_B)$
6:      **end for**
7: **end for**
8: **Return** $B_\theta$

---

### 4.3 Controller Synthesis via Learned Dynamics

The learned barrier function $B$ is subsequently employed to fulfill the primary goal of synthesizing a safe policy 1 using the CBVF-QP formulation in equation 7. Solving this QP necessitates the evaluation of the Lie derivatives $\mathcal{L}_f B$ and $\mathcal{L}_g B$, both of which rely on the underlying control-affine system dynamics. In our offline-only setting, the true dynamics are unavailable; therefore, we construct a neural network–based surrogate model to approximate the underlying transition dynamics of the form:

$$x_{t+1} = x_t + \left( f_\phi(x_t) + g_\phi(x_t) u_t \right) \Delta t, \tag{11}$$

which enables computation of the required derivatives and supports safe policy synthesis. The model parameters $\phi$ are trained using one-step transitions from the offline dataset $\mathcal{D}$ by minimizing the prediction loss:

$$\mathcal{L}_{\text{dyn}}(\phi) = \mathbb{E}_{(x,u,x') \sim \mathcal{D}} \left[ \| x' - \left( x + (f_\phi(x) + g_\phi(x)u)\Delta t \right) \|^2 \right], \tag{12}$$

implemented as a minibatch MSE objective. The full training procedure is summarized in Algorithm 2.

Importantly, the learned dynamics model is *not* used when learning the barrier function. Incorporating it into the CBVF learning stage would require evaluating terms involving $(f_\phi, g_\phi)$ under actions outside the dataset-supported set $\mathcal{U}_\mathcal{D}$, thereby violating the action constraints critical for preventing value underestimation in the offline regime. Hence, using learned dynamics during CBVF training would allow the network to extrapolate into unsupported regions of the action space, defeating the purpose of the OOD-aware barrier learning objective described earlier.

In contrast, at *inference* time, the learned dynamics serve a different role: they enable the evaluation of Lie derivatives needed to solve the CBVF-QP (equation 7). Specifically, for any query state $x$, we compute

$$\mathcal{L}_f B(x) = \nabla_x B_\theta(x)^\top f_\phi(x), \qquad \mathcal{L}_g B(x) = \nabla_x B_\theta(x)^\top g_\phi(x). \tag{13}$$

These quantities allow the QP in equation 7 to be solved for the safe control action $u_{safe}$, completing the pipeline for constructing a safe controller purely from offline demonstrations.

Note that one-step forward-invariance holds under the idealized assumptions (exact barrier and exact Lie-derivatives), and the approximation sources like function approximation, finite-data estimation, and model error can break the closed-loop guarantee in practice. In Section 5.2.2, we provide an analysis demonstrating the limitations of using learned dynamics inside the CBVF learning loop, further reinforcing the necessity of restricting their use to the inference stage only.

## 5 Experiments

The experiments are designed to evaluate: (*i*) the safety and performance of V-OCBF relative to constrained offline RL and neural CBF baselines on systems with unknown dynamics, (*ii*) the advantages of value-guided barriers over behavior-policy–induced barriers, (*iii*) the robustness of the resulting QP controller under external disturbances, and (*iv*) the effectiveness of V-OCBF compared to a CBVF synthesized using learned dynamics.

---

**Algorithm 2** Controller Synthesis

---

**Require:** Offline dataset $\mathcal{D} = \{(x, u, x')\}$, batch size $m$, epochs $N$, learning rate, $\Delta t$, $u_{ref}(\cdot)$, $\kappa$
1: Initialize $\phi$ and optimizer
2: **for** epoch = 1 to $N$ **do**
3:     **for** minibatch $\mathcal{B} \subset \mathcal{D}$ **do**
4:         $L_{\text{dyn}}(\phi) \leftarrow (x, u, x', \Delta t, \phi)$                                ▷ From Eq. 12
5:         $\phi \leftarrow \text{AdamStep}(\phi, \nabla_\phi L_{\text{dyn}})$
6:     **end for**
7: **end for**

8: **Inference at state $\hat{x}$**
9: $L_f B(\hat{x}) \leftarrow (\nabla_x B_\theta(\hat{x}), f_\phi(\hat{x}))$ and $L_g B(\hat{x}) \leftarrow (\nabla_x B_\theta(\hat{x}), g_\phi(\hat{x}))$         ▷ From Eq. 13
10: $u_{safe} \leftarrow \texttt{QP-Solver}\big(L_f B(\hat{x}), L_g B(\hat{x}), u_{\text{ref}}(\hat{x}), B_\theta(\hat{x}), \kappa\big)$         ▷ From Eq. 7
11: **return** safe controller $u_{safe}(\hat{x})$

---

**Baselines:** We compare V-OCBF against a diverse set of constrained offline learning and CBF-based methods. For constrained offline learning, we include **Behavior Cloning (BC)**, **BEAR-Lag** (Lagrangian constraint version of (Kumar et al., 2019)), **COptiDICE** (Lee et al., 2022), and **FISOR** (Zheng et al., 2024) which enforce safety indirectly via soft constraints on policy optimization or behavior imitation. For CBF-based approaches, we evaluate **Neural Control Barrier Function (NCBF)** (Robey et al., 2020), *Conservative Control Barrier Function (CCBF)* (Tabbara & Sibai, 2025), and *In-Distribution Barrier Function (iDBF)* (Castañeda et al., 2023), which synthesize explicit safety filters from offline data but often yield conservative safe sets. In contrast, V-OCBF learns a *value-guided* barrier function from offline demonstrations that accounts for future unsafe interactions, producing a state-wise safety filter that aims to reduce violations and increase safe-set coverage in practice.

**Evaluation Metrics:** We evaluate all methods based on (i) *safety*, measured as the total number of safety violations incurred before episode termination, and (ii) *performance*, measured via the cumulative episode rewards. These metrics allow us to assess the trade-off between strict safety enforcement and task performance across different offline RL and CBF-based approaches.

## 5.1 Experimental Case Studies

To perform a holistic performance analysis of our proposed approach, we apply V-OCBF in conjunction with Behavior Cloning (BC) as the nominal (reference) controller for all the different environments which are supposed to assess varying objectives. Below we list all the environments that we use:

- **Autonomous Ground Vehicle (AGV) Collision Avoidance:** In our first experiment, we examine a 3-dimensional collision avoidance problem involving an autonomous ground vehicle governed by Dubins' car dynamics (Dubins, 1957). The objective is to ensure safety by avoiding a static obstacle while navigating through a bounded environment. Further details on the system dynamics, state space bounds, and experimental setup are provided in Appendix B.1.

- **MuJoCo Safety Gymnasium:** We next evaluate our framework on Safety Gymnasium environments (Ji et al., 2023). Specifically, we evaluate the V-OCBF-based QP (equation 7) on high-dimensional MuJoCo tasks like Hopper, Swimmer, Half Cheetah, Walker2D and Ant. The objective in each environment is to maximize reward while keeping the agent velocity below the velocity thresholds. We keep the reward and safety-violation metrics identical to the Safety-Gymnasium definitions and use the standard DSRL dataset for safe offline RL (Liu et al., 2024). To evaluate our method against baselines, we randomly sampled 500 initial states for each environment, respectively, the results for which can be referred to from Figure 2.

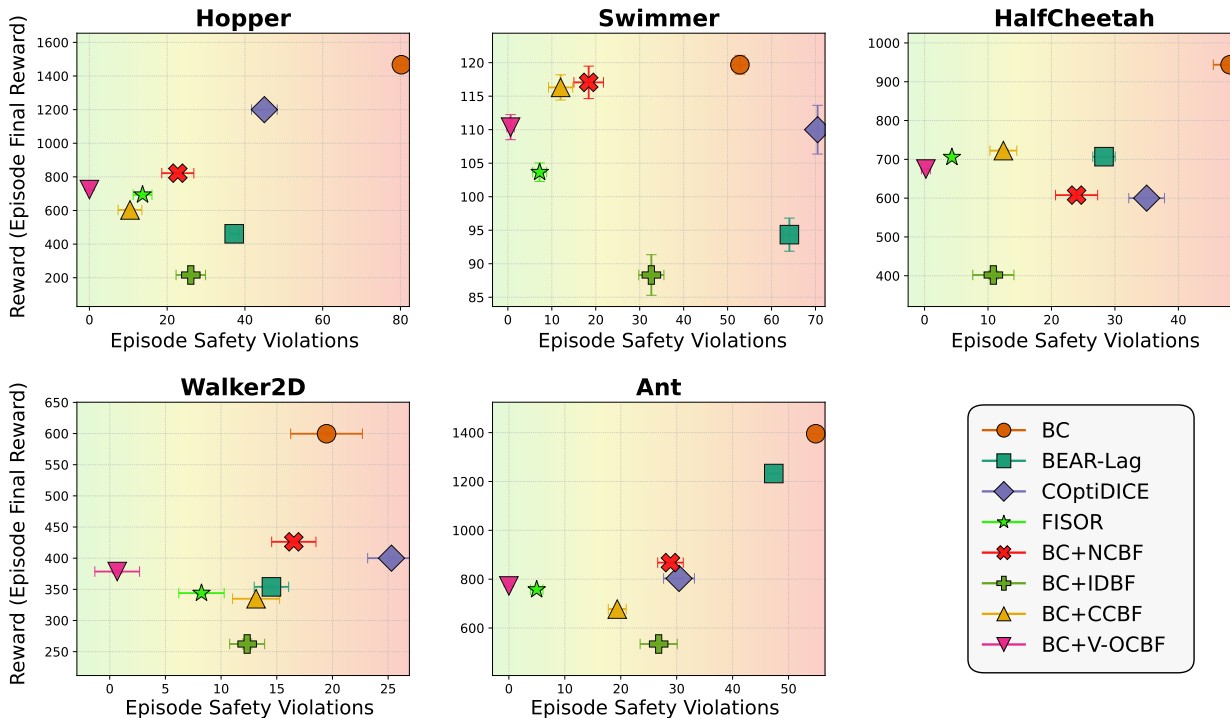

Figure 2: Results plot for all the Mujoco based Safety Gymnasium (Ji et al., 2023) environments. Points towards left (←) are more safe than those on right (→). Evaluated over 500 episodes and 5 seed values.

## 5.2  Results

### 5.2.1  Effectiveness in Co-optimizing Safety and Performance

We begin by evaluating all methods on the AGV Collision Avoidance task, which provides a clear setting to study how different approaches balance safety and performance. The results in Table 1 highlight notable differences in how offline RL and CBF-based methods handle this trade-off.

Offline RL baselines such as BC, BEAR-Lag, and COptiDICE achieve relatively low safety rates. BC tends to reproduce unsafe behaviors from the dataset, while BEAR-Lag and COptiDICE try to account for safety but remain limited because they operate with soft constraint formulations. Their lower reward and safety scores indicate that they struggle to balance both safety and performance objectives.

In contrast, methods that incorporate a CBF-QP layer, such as BC+NCBF, BC+iDBF, and BC+CCBF, achieve much higher safety rates. The QP ensures that unsafe actions are filtered out, even if the nominal controller is imperfect. However, the performance of these approaches still depends heavily on the quality of the learned barrier. FISOR performs better than the other offline RL baselines because it explicitly expands the feasible safe region before optimizing for performance. However, due to lack of explicit safety filtering, it leads to lesser safety rates than our proposed method, due to the impending learning errors in the computation of feasible region. This also highlights the importance of QP based safety filtering scheme for achieving better safety. Overall, *V-OCBF achieves the strongest results across all metrics.*

To further analyze scalability, we extend the evaluation to MuJoCo Safety Gymnasium environments (Hopper, Half-Cheetah, Ant, Swimmer, and Walker2D), with unknown dynamic models. The results in Figure 2 demonstrate that V-OCBF again achieves the lowest safety violation rates across all tasks. Notably, the method maintains near-zero violations on while preserving satisfactory reward levels compared to BC and outperforming iDBF, NCBF, and CCBF. These neural CBF baselines degrade sharply in higher dimensions: NCBF suffers from optimization difficulties, while iDBF often enforces overly restrictive boundaries that

Table 1: AGV Collision Avoidance Experiment: Percentage Safe Episodes, Mean Episode Reward and Safe Set Volume (refer Appendix B.1.1) across different methods. Evaluated over 500 episodes and 5 seed values.

| Method | Safe Episodes (%) | Episode Reward | Safe Set Volume (%) |
|---|---|---|---|
| BC | $48.92 \pm 1.69$ | $20.45 \pm 1.84$ | 42.51 |
| BEAR-Lag (Kumar et al., 2019) | $65.12 \pm 0.24$ | $13.85 \pm 0.81$ | 58.21 |
| COptiDICE (Lee et al., 2022) | $68.91 \pm 0.32$ | $15.33 \pm 0.67$ | 62.32 |
| BC+NCBF (Robey et al., 2020) | $92.48 \pm 0.60$ | $44.61 \pm 2.58$ | 81.92 |
| BC+iDBF (Castañeda et al., 2023) | $92.87 \pm 0.73$ | $48.23 \pm 2.01$ | 83.32 |
| BC+CCBF (Tabbara & Sibai, 2025) | $93.56 \pm 0.56$ | $49.66 \pm 2.34$ | 90.94 |
| FISOR (Zheng et al., 2024) | $95.78 \pm 0.2$ | $52.33 \pm 0.93$ | 90.14 |
| BC+**V-OCBF** (Ours) | $\mathbf{98.28 \pm 0.54}$ | $\mathbf{54.93 \pm 0.46}$ | **92.57** |

suppress task performance. FISOR again remains competitive but does not match the safety consistency of V-OCBF.

Overall, the experiments provide strong empirical evidence that V-OCBF effectively co-optimizes safety and performance, scaling from low-dimensional AGV dynamics to complex MuJoCo systems. The method consistently outperforms existing offline RL and neural CBF baselines in terms of safety while maintaining competitive reward, highlighting its suitability for offline settings where both strict safety and reliable performance are required.

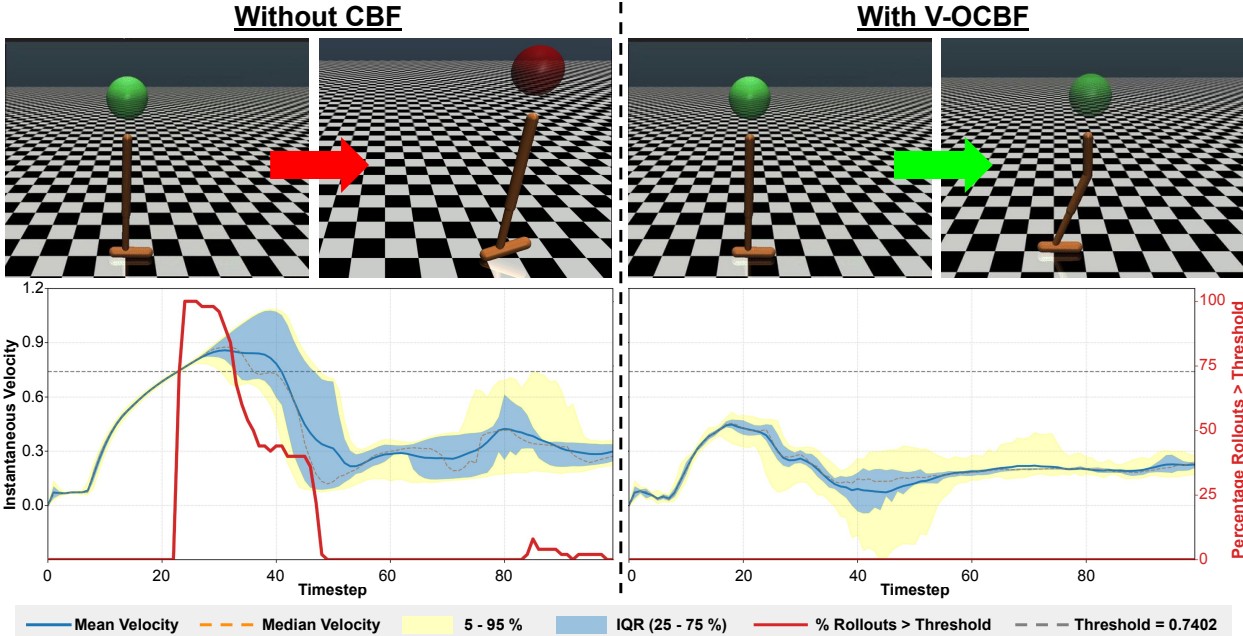

Figure 3: **Qualitative comparison of Hopper rollouts with and without V-OCBF.** *(Top):* When using V-OCBF, the agent adapts its gait to keep its forward velocity within the allowable safety threshold, demonstrating how the learned barrier actively regulates unsafe accelerations. *(Bottom):* Instantaneous velocities across episodes for both methods. The left axis reports the velocity profile, while the right axis shows the fraction of episodes that violate the safety threshold at each timestep. V-OCBF substantially reduces violation frequency, illustrating its ability to enforce safety constraints during deployment. Additional qualitative results for other environments are provided in Appendix B.2.1.

### 5.2.2 Effect of learned dynamics

We additionally examine how using the learned dynamics influences both CBVF learning and QP-based control on the AGV environment, where ground-truth dynamics are available for comparison. Table 2 reports

safety and reward for three configurations: (i) V-OCBF with a QP evaluated using learned dynamics, (ii) V-OCBF with a QP evaluated using true dynamics, and (iii) a neural CBVF trained using learned dynamics and QP-based controller evaluated on the same dynamics.

Across all metrics, using learned dynamics at inference time yields performance that is close to the known-dynamics case, indicating that the controller synthesis using QP is robust to moderate model error. In contrast, using learned dynamics during CBVF training leads to a noticeable drop in both safety and reward. This aligns with our earlier discussion in section 4.3: learning the barrier through a model introduces distributional mismatch because the learned dynamics may generate actions outside the dataset-supported set $\mathcal{U}_{\mathcal{D}}$, yielding suboptimal targets for $B$. These findings reinforce our design choice, dynamics should be used only at inference to compute the lie derivatives in the QP, while barrier learning itself should avoid dependence on a learned model.

Table 2: Use of Learned Dynamics at the time of Training v/s Inference (Mean over 500 episode rollouts).

| Method | Safe Episodes (%) | Episode Reward |
|---|---|---|
| BC+**V-OCBF** (QP with Learned Dynamics) | $98.28 \pm 0.54$ | $54.93 \pm 0.46$ |
| BC+**V-OCBF** (QP with Known Dynamics) | $99.52 \pm 0.58$ | $55.91 \pm 0.39$ |
| BC+**CBVF from Learned Dynamics** | $94.64 \pm 0.65$ | $46.40 \pm 0.02$ |

### 5.3 Ablation Studies and Additional Experiments

To further evaluate the robustness of V-OCBF, we conduct a series of studies that examine: (i) the sensitivity of the learned safe set to the size of the barrier network, (ii) the effect of disturbances on the safety of QP based controller, (iii) the dependence of the learned barrier on the expectile parameter $\tau$, and (iv) the resilience of the framework to approximation errors in the learned dynamics model. Additionally, we extend our evaluation to a boat navigation task with nonlinear drift to assess performance under complex environmental dynamics. Across these experiments, V-OCBF exhibits stable safety performance over a broad range of architectural and hyperparameter choices, remaining resilient even under perturbations to the control actions. These findings suggest that the framework generalises reliably beyond the specific tasks considered and is well-suited for settings where safety certification must be carried out under imperfect system knowledge. The full results and visualisations for these studies are presented in Appendix C and Appendix D.

## 6 Conclusion, Limitations and Future works

This work presented V-OCBF, a value-guided framework for learning control barrier functions directly from offline demonstrations. By combining a finite-difference CBVF recursion with an expectile-based objective, V-OCBF propagates safety information in a principled manner while avoiding evaluations on unsupported actions, enabling reliable CBF-QP control even when dynamics are unknown and action authority is limited. Experiments across AGV and high-dimensional Safety Gymnasium environments show that V-OCBF consistently reduces safety violations while maintaining strong task performance, outperforming both constrained offline RL and neural CBF baselines.

While V-OCBF reliably learns useful control-barrier candidates from offline demonstrations and enables QP-based safety filtering under unknown dynamics and limited action authority, the learned barrier is not a formally verified certificate, function approximation and learned-dynamics errors mean guarantees remain empirical. Furthermore, the framework is not explicitly trained to handle adversarial disturbances or worst-case model errors. To close this gap, future work will explore integration of formal-certification (e.g., Lipschitz-based verification (Tayal et al., 2024b) and conformal prediction (Lindemann et al., 2025)). Moreover, we will explore robust extensions of V-OCBF that incorporate adversarial or uncertainty-aware training objectives, enabling safety guarantees under stronger disturbance models and broader distribution shifts.

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

# A Theoretical Insights

## A.1 Finite Difference Barrier Condition

**Lemma 1** (Finite-Difference Approximation of the Barrier Condition). *Consider a control-affine system $\dot{x} = f(x) + g(x)u$ with a continuously differentiable barrier function $B : \mathcal{X} \to \mathbb{R}$. The continuous-time barrier condition*

$$\min\{ \max_u \nabla B(x)^\top (f(x) + g(x)u), \; \ell(x) - B(x) \} = 0 \tag{14}$$

*admits the finite-difference approximation*

$$\min\{ \max_{u_t} B(x_{t+1}), \; \ell(x_t) \} = B(x_t), \tag{15}$$

*up to a positive scaling by $\Delta t$.*

*Proof.* Starting from the continuous-time condition in equation 14, note that

$$\nabla B(x)^\top (f(x) + g(x)u) = \nabla B(x)^\top \dot{x} = \frac{d}{dt} B(x(t)). \tag{16}$$

Using a forward Euler approximation, the time derivative satisfies

$$\frac{d}{dt} B(x(t)) \approx \frac{B(x_{t+1}) - B(x_t)}{\Delta t}, \tag{17}$$

where, $x_t = x(t)$, $u_t = u(t)$ and $x_{t+1} = x(t + \Delta t)$. Substituting equation 17 into equation 14 yields

$$\min \left\{ \max_{u_t} \frac{B(x_{t+1}) - B(x_t)}{\Delta t}, \; \ell(x_t) - B(x_t) \right\} = 0. \tag{18}$$

Let $a = B(x_{t+1}) - B(x_t)$ and $b = \ell(x_t) - B(x_t)$. Since $\Delta t > 0$, the function

$$m(a, b) = \min\{ a/\Delta t, \; b \}$$

satisfies $m(a, b) = 0$ if and only if $\min\{a, b\} = 0$. This follows from the fact that scaling one argument of the minimum by a positive constant does not change which argument is smaller, nor the value at which the minimum equals zero. Thus, equation 18 is equivalent to

$$\min\{ a, \; b \} = 0, \tag{19}$$

which gives the finite-difference barrier condition in equation 15. This establishes the claim. $\qquad\square$

## A.2 Expectile Regression

Expectile regression is a classical tool in statistics and econometrics for estimating asymmetric statistics of a random variable. For a random variable $X$, the $\tau$-expectile is defined as the minimizer of the asymmetric least-squares problem

$$m_\tau = \arg\min_m \mathbb{E}_{x \sim X} \left[ L^\tau(x - m) \right], \tag{20}$$

where $L_\tau(y) = |\tau - \mathbf{1}(y < 0)|, y^2$.

When $\tau > 0.5$, this loss assigns more weight to samples above the estimate $m_\tau$ and less weight to samples below it. Conversely, $\tau < 0.5$ emphasizes lower values. Thus, expectiles interpolate smoothly between the mean ($\tau = 0.5$) and a "high-value–seeking" statistic as $\tau \to 1$.

Expectile regression can also be extended to learning conditional expectiles:

$$m_\tau(x) = \arg\min_{m(\cdot)} \mathbb{E}_{(x,y) \sim \mathcal{D}} \left[ L^\tau(y - m(x)) \right], \tag{21}$$

which can be optimized efficiently via stochastic gradient descent. This makes expectiles easy to implement in modern machine-learning pipelines, unlike alternative high-order statistics that require specialized solvers.

### A.2.1  Why Expectile Regression in V-OCBF?

In V-OCBF, the barrier function ideally requires computing $\max_{u' \in \mathcal{U}} B(x')$, but performing this maximization directly is problematic in the offline setting because the dataset supports only a restricted action set $\mathcal{U}_\mathcal{D} = \{ u \mid (x, u, x') \in \mathcal{D} \}$, and querying $B(x')$ for unseen actions $u' \notin \mathcal{U}_\mathcal{D}$ introduces unsupported targets that can distort or destabilize learning. Moreover, even restricting the maximization to $\mathcal{U}_\mathcal{D}$ leads to a brittle update since a strict maximum is extremely sensitive to noise or rare outlier transitions, often producing poor barrier estimates. Expectile regression provides a smooth and data-supported alternative by emphasizing higher (safer) barrier values when $\tau > 0.5$, effectively approximating the maximal barrier value over $\mathcal{U}_\mathcal{D}$ while never querying out-of-distribution actions, thereby yielding a stable and principled mechanism for synthesising offline CBVF.

**Expectile regression resolves these issues.** For $\tau > 0.5$, the expectile estimator concentrates on the largest barrier values consistent with the dataset. This yields a smooth approximation to $\max_{u' \in \mathcal{U}_\mathcal{D}} B(x')$, which is equivalent to performing a constrained maximization over only the dataset-supported actions, without ever querying $B$ on out-of-distribution actions.

To illustrate the effect of the expectile parameter we show barrier functions learned for the AGV case study using $\tau \in \{0.5, 0.6, 0.7, 0.8, 0.9, 0.99\}$ (Figure 4). As $\tau$ increases the volume of the sub-zero level set visibly shrinks: lower $\tau$ values produce larger sub-zero regions because the learning target is more influenced by poorer next-state outcomes present in the data, which inflates the set of states that appear unsafe under the learned barrier. By contrast, larger $\tau$ up-weights higher (safer) barrier targets within the dataset and yields a smoother, upper-envelope–like approximation to the maximization over dataset-supported actions; sampling actions from this higher expectile therefore produces a smaller sub-zero level set (i.e., a tighter unsafe region) and correspondingly larger estimated safe set.

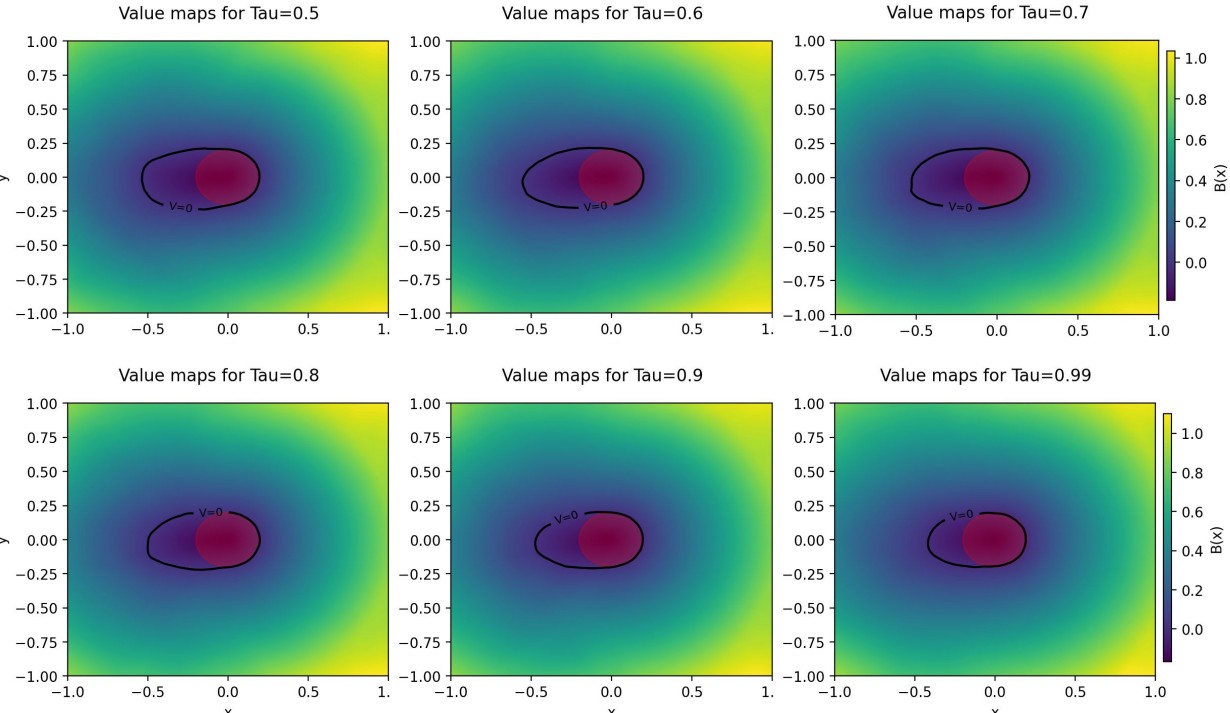

Figure 4: $\tau$-**Expectiles:** Figure shows a depiction of the learned Barrier Function for different $\tau$ values with Expectile Regression. With increasing value of $\tau$ ($> 0.5$), the feasible region increases, indicating a more accurate representation of the true barrier function.

### A.3 Feasibility of V-OCBF-QP

Formal safety guarantees of standard QP-based CBF formulations generally hold under the assumption of unbounded actuation. In practice, however, in the presence of control bounds, a nominal CBF may conflict with feasibility requirements, causing the corresponding CBF-QP to become infeasible. To address this within our framework, we adopt a standard relaxation technique from the safe control literature by introducing a slack variable $\delta$. This relaxes the hard safety constraint when strictly necessary while heavily penalizing any violations. The resulting slack-augmented V-OCBF-QP is formulated as:

$$\pi_{\text{safe}}(x) = \arg\min_{u \in \mathbb{U}, \delta \in \mathbb{R}} \|u - \pi_{\text{ref}}(x)\|^2 + \lambda \delta^2$$
$$\text{s.t. } \mathcal{L}_f B(x) + \mathcal{L}_g B(x) u + \kappa\left(B(x)\right) + \delta \geq 0,$$

(22)

where $\lambda > 0$ is a penalty coefficient.

Empirically, $\delta$ remains close to zero throughout our experiments. Because V-OCBF explicitly accounts for control constraints during learning, the synthesized policy naturally respects admissible bounds. Consequently, the safety filter rarely becomes infeasible, and any triggered corrective adjustments require minimal slack usage.

## B  Description of the Experiments

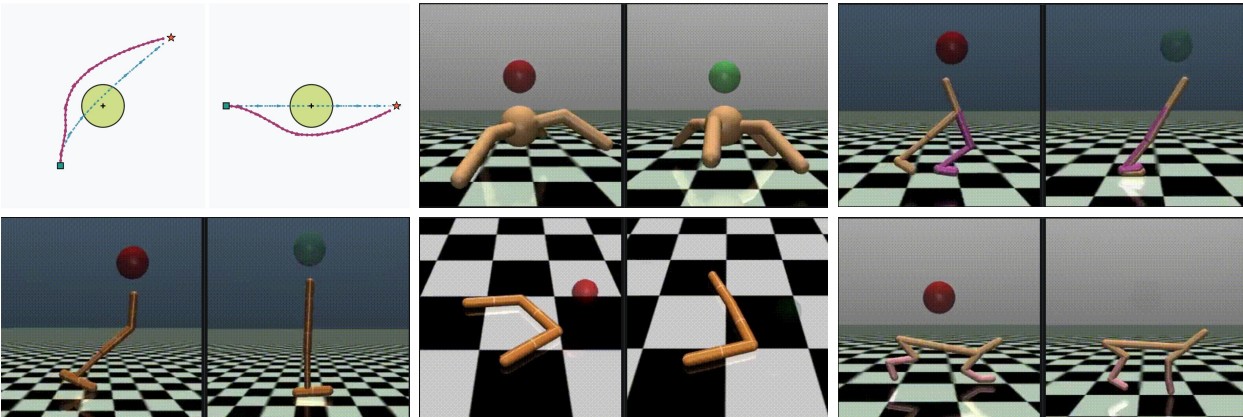

Figure 5: **Illustrations of the test environments:** (*Top-Left*) environment illustrates Autonomous Ground Vehicle with an obstacle and a goal point. (*MuJoCo Env*) **Red** sphere denotes unsafe state.

### B.1  Autonomous Ground Vehicle Collision Avoidance

We consider a collision-avoidance task for an autonomous ground vehicle modeled using the Dubins dynamics:

$$\begin{bmatrix} \dot{x}_1 \\ \dot{x}_2 \\ \dot{\Phi} \end{bmatrix} = \begin{bmatrix} v \cos \Phi \\ v \sin \Phi \\ 0 \end{bmatrix} + \begin{bmatrix} 0 \\ 0 \\ 1 \end{bmatrix} u,$$

(23)

where the state $x = [x_1, x_2, \phi]^T \in \mathcal{X} \subseteq \mathbb{R}^3$ encodes the vehicle's position and heading. The forward velocity is fixed at $v = 0.6$ units/s, and the admissible state space is given by $\mathcal{X} = [-1, 1]^2 \times [-\pi, \pi]$. The control input $u \in [-1, 1]$ denotes the angular velocity $\omega$.

The reward function $r(x)$ is defined as

$$r(x) = \frac{C}{\left( \left\| \begin{bmatrix} x_1 \\ x_2 \end{bmatrix} - \begin{bmatrix} x_{g1} \\ x_{g2} \end{bmatrix} \right\| + \epsilon \right)} \tag{24}$$

where $C = 0.1$ is the scaling factor and $(x_{g1}, x_{g2})^T$ denotes the goal location. Rewards are accumulated only while the agent remains collision-free. The constant $\epsilon > 0$ is included to prevent numerical instability when the vehicle approaches the goal.

Safety is encoded through the function

$$\ell(x) = \left\| \begin{bmatrix} x_1 \\ x_2 \end{bmatrix} - \begin{bmatrix} x_{o1} \\ x_{o2} \end{bmatrix} \right\| - R, \tag{25}$$

where $(x_{o1}, x_{o2})^T = (0,0)^T$ denotes the center of the obstacle and $R = 0.2$ units is the radius of the obstacle.

**Offline Data Generation.**   Since this is a custom environment, we construct an offline dataset for training both the barrier function and the safe policy. We sample 1500 initial states uniformly from $\mathcal{X}$ and simulate each trajectory for 500 discrete timesteps with step size $\Delta t = 0.01s$. During data collection, control inputs are drawn uniformly at random from the admissible range, ensuring diverse system trajectories for learning-based safety certification.

### B.1.1   Safe Set Volume Analysis

Backward Reachable Tube is the set of initial states for which the agent acting optimally, will eventually reach the target set $\mathcal{T}$ within the time horizon $[0, t]$:

$$SSV(x) = \{x : \exists u(\cdot), \exists t \in [0, T], \xi_{x,0}^u(t) \in \mathcal{T}\} \tag{26}$$

where $\xi_{x,0}^u(t)$ represents system state at time $t \in [0, T]$, and $u$ defined over time horizon $[0, T]$, represents sequence of control actions over the time horizon.

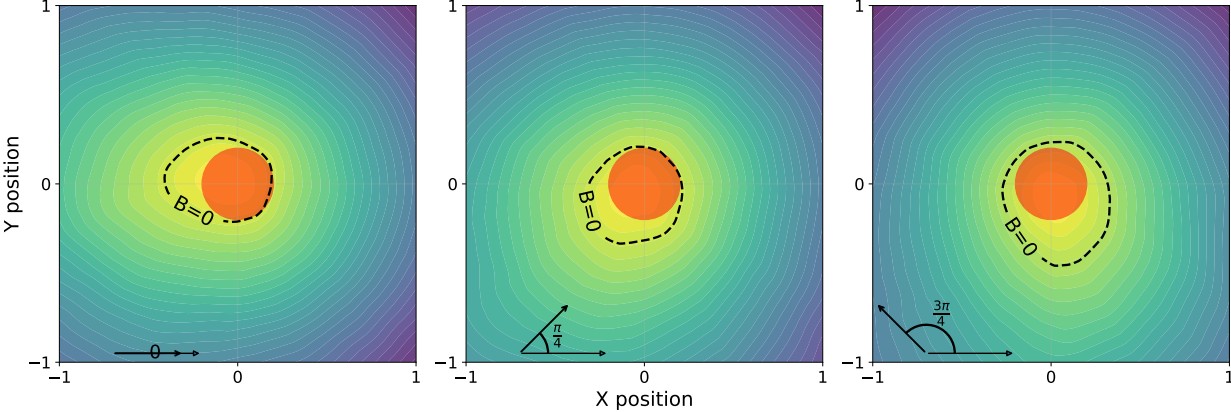

Figure 6: **Safe Set Volume:** Figure shows a depiction of feasible region (i.e., anything outside B=0 enclosure) for V-OCBF for 3 different heading angles. Note that calculation of the actual Safe Set volume is a cumulative of feasible region over countless such frames along the continuous heading angle ($\phi$) dimension.

We performed Safe Set Volume (SSV) analysis to inspect the effectiveness and conservatism of various approaches. Specifically, we aim to evaluate the volume of the state space from which the agent can safely initialize. Effectively, the larger this Safe Set volume is, bigger the feasible region gets, hence, making it easy to navigate in the environment, and smaller the Safe Set volume becomes, lesser the feasible region gets, thus, overall making it less safe for the agent to navigate in the environment.

To evaluate the safe initialization region that each of the method casts, we uniformly sample different initial states across the state space and check whether they belong to the above-defined Safe Set Volume by rolling out trajectories in the online environment for over 100k times (as we started to observe convergence in the Safe Set Volume after the 100k mark). We investigate the results for this analysis in the main paper (refer Table 1) where we show that V-OCBF enables the largest safe feasible region among all the various approaches.

## B.2 Safety MuJoCo Environments:

To evaluate our framework on higher-dimensional systems, we use MuJoCo Safety Gymnasium environments Ji et al. (2023), where safe velocity tasks introduce velocity constraints for Gymnasium's MuJoCo-v4 agents.

The agent receives a scalar cost signal on the basis of its velocity at each step:

$$cost = bool(V_{current} > V_{threshold}) \tag{27}$$

This constraint can be framed as a safety function

$$\ell(x) = V_{threshold} - V_{current} \tag{28}$$

since $\ell(x) \geq 0$ corresponds to safe state. This safety value function provides us a continuous signal instead of sparse values of 0 or 1. For agent specific threshold velocities and time-step $\Delta t$, we refer the official documentation, the official values for which have been compiled and included in Table 3.

Table 3: Threshold velocity $V_{threshold}$, time step size $\Delta t$ and action space $\mathbb{U}$ for each MuJoCo based Safety Gymnasium Ji et al. (2023) environment as per official documentation.

| Environment | Hopper | Half-Cheetah | Ant | Walker2D | Swimmer |
|---|---|---|---|---|---|
| $V_{threshold}$ | 0.7402 | 3.2096 | 2.6222 | 2.3415 | 0.2282 |
| $\Delta t$ | 0.008 | 0.05 | 0.05 | 0.008 | 0.04 |
| $\mathbb{U}$ | $[-1.0, 1.0]^3$ | $[-1.0, 1.0]^6$ | $[-1.0, 1.0]^8$ | $[-1.0, 1.0]^6$ | $[-1.0, 1.0]^2$ |

### B.2.1 Qualitative Analysis

In continuation to the Hopper plot in the main paper (Fig. 3), we include plots for all the remaining MuJoCo safety gymnasium environments covered in this paper with the visualization of the episode rollouts, both with and without V-OCBF. Given the safety rates of V-OCBF compared to the other baselines (refer Fig. 2), it becomes important to visually realize the real-time effect on performance (Figures 7, 8, 9).

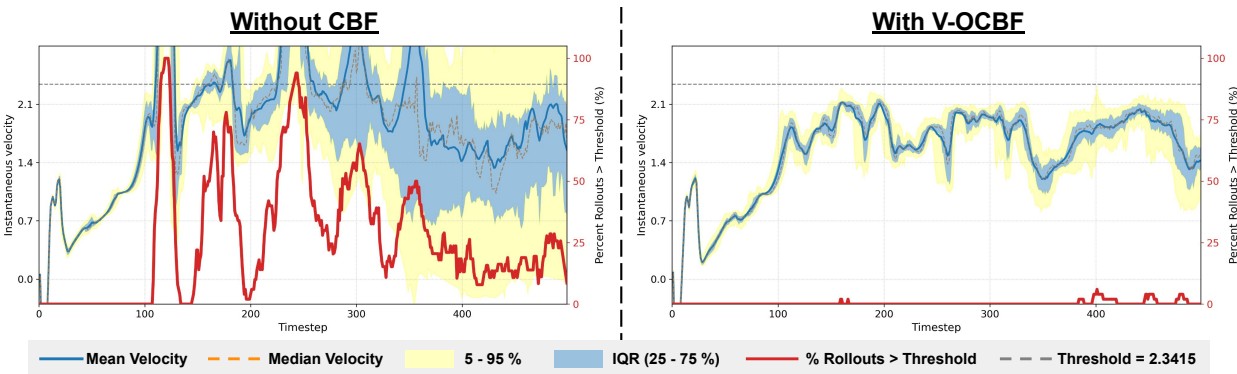

Figure 7: **Walker-2D.** Instantaneous velocity (left Y-axis) across episodes. Right Y-axis depicts percentage of episodes when the agent violates the velocity threshold at a particular timestep.

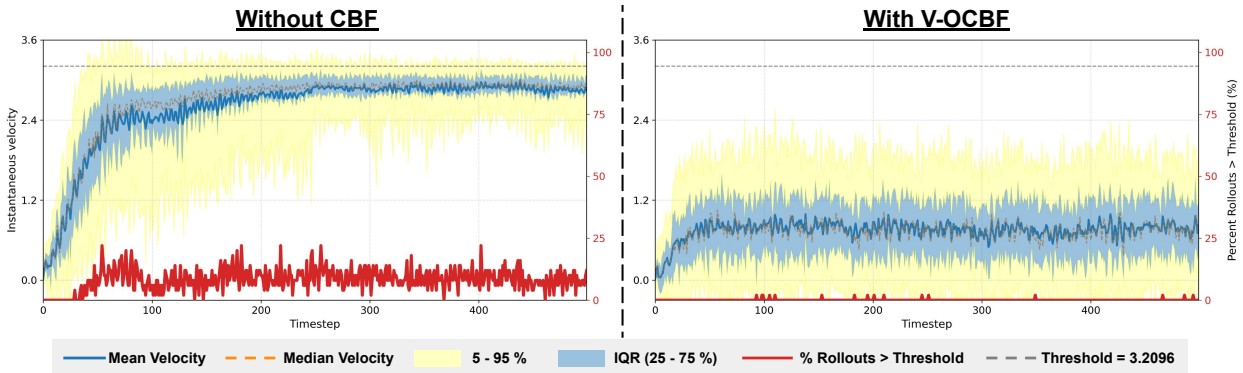

Figure 8: **Half-Cheetah.** Instantaneous velocity (left Y-axis) across episodes. Right Y-axis depicts percentage of episodes when the agent violates the velocity threshold at a particular timestep.

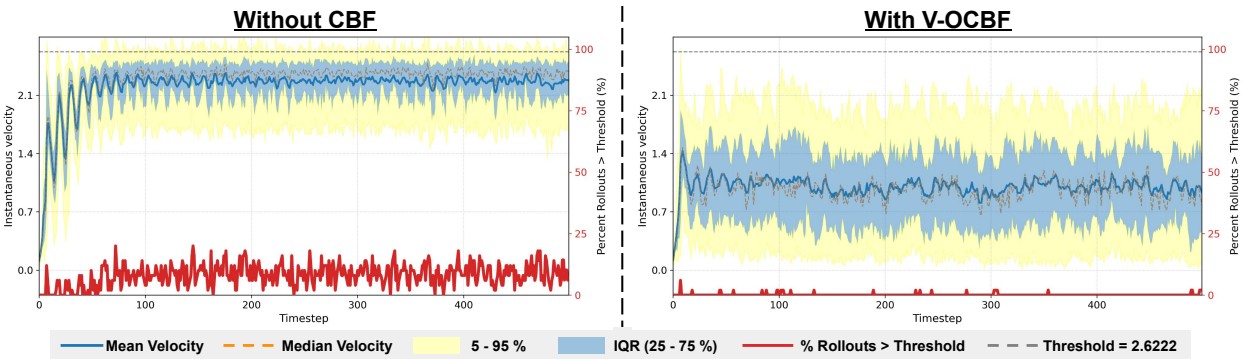

Figure 9: **Ant.** Instantaneous velocity (left Y-axis) across episodes. Right Y-axis depicts percentage of episodes when the agent violates the velocity threshold at a particular timestep.

# C   Ablation Studies

We have performed a set of ablation studies on our primary testing environment of Autonomous Ground Vehicle Collision Avoidance to further strengthen our claims of the paper. AGV acts as an ideal testing bed for such ablations as it is easy to visualize and infer from the results in a 2D space. Besides, the environment properly encapsulates our requirements of providing a safety critical control environment. Following subsections cover the ablations we have performed.

## C.1   Robustness to size of the NN

In this section we show our framework's robustness against varying model sizes, thus, demonstrating the versatility on different hyperparameter choices. Table 4 shows the safety rate for different model dimensions.

Table 4: Safety Rate (%) reported for various model dimensions. Columns represent Width: neurons in each hidden layer, and Rows represent number of layers.

| Width →
Layers ↓ | 256
Neurons | 96
Neurons | 64
Neurons | 48
Neurons |
|---|---|---|---|---|
| 2 | **98.28 ± 0.54** | 97.18 ± 0.41 | 96.96 ± 1.21 | – |
| 3 | 98.04 ± 0.30 | 97.36 ± 0.22 | 97.12 ± 0.23 | 96.54 ± 1.62 |
| 4 | – | – | – | 96.82 ± 1.24 |

As can be seen from the Table 4, the models don't affect the safety performance significantly, until we drop the size by too much (as in the cases of **(48,3)**, **(48,4)** and **(64,2)**). We consider the subtle drop is safety rate for the lower dimensional networks to be within considerable limits.

## C.2  Robustness to errors in Learned Dynamics

To evaluate the robustness of our pipeline to errors in the learned dynamics, we measure how additive perturbations affect the safety performance of the CBF-based controllers. Concretely, we perturb the computed safe action $\omega$ by adding zero-mean Gaussian noise $\mathcal{N}(0, \sigma^2)$ with a range of standard deviations $\sigma$ (refer Table 5). For every noise level we run 500 independent trials (with 5 different random seeds each) and report the safety rate defined as the fraction of trials that remain within the safe set over the evaluation horizon. Table 5 summarizes the results across several CBF variants, for each method we report mean safety rate $\pm$ standard error. Across noise levels spanning small to large perturbations, the safety rates degrade only mildly, indicating that the learned CBFs tolerate moderate inaccuracies in the dynamics / control signal. These results suggest that, while precise dynamics models improve performance, our approach provides substantially safety-aware control even when the dynamics model is imperfect.

Table 5: Safety Rate (%) reported for Robustness Testing of **learned Dynamics** for Autonomous Ground Vehicle by adding upto 5%, 10% and 20% of the possible control action as noise.

| Method | No Noise | + Small Noise (5%) | + Medium Noise (10%) | + Large Noise (20%) |
|---|---|---|---|---|
| BC+NCBF Robey et al. (2020) | $92.48 \pm 0.60$ | $92.48 \pm 0.62$ | $92.52 \pm 0.41$ | $92.32 \pm 0.56$ |
| BC+iDBF Castañeda et al. (2023) | $92.87 \pm 0.73$ | $92.84 \pm 0.64$ | $92.40 \pm 0.84$ | $92.76 \pm 0.28$ |
| BC+CCBF Tabbara & Sibai (2025) | $93.56 \pm 0.56$ | $93.72 \pm 0.58$ | $93.68 \pm 0.80$ | $93.76 \pm 0.38$ |
| BC+**V-OCBF** (Ours) | $\mathbf{98.28 \pm 0.54}$ | $\mathbf{98.08 \pm 0.61}$ | $\mathbf{97.84 \pm 0.68}$ | $\mathbf{97.94 \pm 0.48}$ |

This empirical test shows that even if we haven't learned the most accurate dynamics model, we can expect a considerably precise safety-aware control for the system with our approach.

## C.3  Sensitivity to Expectile parameter ($\tau$)

As discussed in Section 4.2, we use a $\tau$-expectile loss to obtain the final CBVF. When $\tau = 0.5$, the objective reduces to a symmetric MSE fit, causing the learned barrier to reflect the average safety value induced by the behavior policy. This reproduces the *behavior-induced* barrier and typically yields overly conservative safe sets that do not generalize well beyond the demonstrated trajectories.

To investigate this effect, we conduct a study by comparing $\tau = 0.5$ against higher expectile levels that emphasize the upper tail of the safety value distribution. Table 6 summarises the safety rates achieved on the AGV environment for different $\tau$ values.

Table 6: Safety Rate (%) reported for Autonomous Ground Vehicle by varying $\tau$ in the expectile regression.

| $\tau$ | 0.5 (Behavior Policy) | 0.7 | 0.8 | 0.9 | 0.99 |
|---|---|---|---|---|---|
| **Safety Rate (%)** | $96.53 \pm 0.23$ | $96.87 \pm 0.54$ | $97.16 \pm 1.01$ | $97.72 \pm 0.88$ | $\mathbf{98.28 \pm 0.54}$ |

The results reveal two key patterns. First, the behavior-induced barrier ($\tau = 0.5$) yields the lowest safety rate, consistent with its tendency to underrepresent safety-critical regions that are not frequently explored by the behavior policy. Second, as $\tau$ increases, the barrier becomes progressively more conservative in safety-critical areas and better aligned with the upper end of the demonstrated safety values, resulting in consistently higher safety rates. Figure 4 depicts this nature of the learned barrier functions on varying the value of $\tau$.

Under idealized conditions (deterministic dynamics and no dataset disturbances ,i.e., when the best actions reliably lead to the best next states), pushing the expectile $\tau \to 1$ concentrates the estimator on the upper tail of the target distribution and, under favorable (nearly deterministic) conditions, can yield the strong empirical performance, prioritizing high-value next-state targets. However, this extreme setting is brittle to rare poor actions or stochastic disturbances that produce optimistic next states; in such cases an estimator that strictly targets the extreme upper tail may overfit those spurious outcomes.

To empirically validate this reasoning, we conducted an additional set of experiments evaluating the impact of $\tau$ across multiple Safety MuJoCo tasks using standard, public datasets from DSRL, which are not strictly deterministic. We evaluated both safety (measured via Cost) and task performance (measured via Reward) for $\tau \in \{0.5, 0.7, 0.9, 0.99\}$, as shown in Table 7.

Table 7: Reward and Cost across different $\tau$ expectile values for selected MuJoCo environments

| Environment | $\tau = 0.5$ | | $\tau = 0.7$ | | $\tau = 0.9$ | | $\tau = 0.99$ | |
| | Reward | Cost | Reward | Cost | Reward | Cost | Reward | Cost |
| --- | --- | --- | --- | --- | --- | --- | --- | --- |
| Ant | 565.36 | 0.0 | 639.61 | 0.0 | 791.77 | 0.0 | 734.66 | 0.02 |
| Half Cheetah | 100.39 | 0.0 | 180.87 | 0.0 | 679.73 | 0.0 | 582.58 | 0.05 |
| Hopper | 96.94 | 0.0 | 126.54 | 0.0 | 769.69 | 0.0 | 526.72 | 0.27 |

As observed in Table 7, increasing $\tau$ toward 1 (specifically $\tau = 0.99$) results in degraded safety performance, reflected by increased Cost, and in some cases reduced Reward. This trend is consistent across the reported tasks and aligns with our theoretical intuition regarding sensitivity to stochastic transitions. In contrast, $\tau = 0.9$ achieves a more favorable balance between reward maximization and safety preservation. Similar patterns were observed across other environments. Based on these findings, we selected $\tau = 0.9$ for the experiments reported in the main paper.

## C.4 Sensitivity to Dynamics Model Accuracy

To empirically evaluate the resilience of our framework against approximation errors in the learned dynamics, we conducted an analysis using the Autonomous Ground Vehicle environment to quantify how variations in model fidelity influence safety performance. We evaluated four distinct configurations representing varying levels of accuracy: the known system dynamics, the standard learned model used in our main experiments, and two degraded models trained for only 50% and 25% of the standard steps, respectively. We quantified the model quality by computing the mean squared prediction error over a set of randomly sampled state-action pairs, which confirmed a monotonic reduction in predictive accuracy across the models.

Table 8: Effect of Dynamics Model Accuracy on Safety Rates at the time of Inference (Mean over 500 episode rollouts).

| Method | Safe Episodes (%) | Episode Reward | Mean L2 Error |
| --- | --- | --- | --- |
| BC+**V-OCBF** (Known Dynamics) | $99.52 \pm 0.58$ | $55.91 \pm 0.39$ | 0 |
| BC+**V-OCBF** (Learned Dynamics) | $98.28 \pm 0.54$ | $54.93 \pm 0.46$ | $1.48 \times 10^{-2}$ |
| BC+**V-OCBF** (Half-Learned Dynamics) | $97.03 \pm 0.45$ | $52.44 \pm 0.18$ | $2.93 \times 10^{-2}$ |
| BC+**V-OCBF** (Quarter-Learned Dynamics) | $96.32 \pm 0.73$ | $51.91 \pm 0.32$ | $4.46 \times 10^{-2}$ |

Subsequently, we utilized each model to synthesize the controller and evaluated the resulting empirical safety rates. As detailed in Table 8, the results indicate that the safety rate experiences only a modest reduction despite the increase in model prediction error. These findings suggest that V-OCBF maintains its effectiveness in mitigating safety violations even when the underlying dynamics model is imperfect, supporting the practical applicability of the method in settings where exact system identification is challenging.

# D  Additional Experiments on Boat Navigation

To further assess the generalization capabilities of V-OCBF, we evaluated the method on a boat navigation task characterized by nonlinear river drift and static obstacles. This environment, while low-dimensional, presents a control challenge due to the drift dynamics which can actively push the agent into unsafe regions if not properly anticipated. Such drift dynamics are particularly difficult for offline safe RL and CBF-based methods to handle when relying on learned models or indirect constraint formulations.

The system dynamics are given by:

$$(x_1)_{t+1} = (x_1)_t + (a_1 + 2 - 0.5x_2^2) \cdot \Delta t$$
$$(x_2)_{t+1} = (x_2)_t + a_2 \cdot \Delta t$$

where $\Delta t$ represents discrete time step, $a_1, a_2$ represent the velocity control action in $x_1$ and $x_2$ directions respectively, subject to the actuation limit $a_1^2 + a_2^2 \leq 1$. The term $2 - 0.5x_2^2$ introduces a state-dependent drift along the $x_1$-axis. The safety constraints are encoded as:

$$\ell(x) := min(\|x - (-0.5, 0.5)^T\| - 0.4, \|x - (-1.0, -1.2)^T\| - 0.4) \tag{29}$$

where $\ell(x) < 0$ indicates that the boat is inside a obstacle, thereby ensuring that the sub-level set of $\ell(x)$ defines the failure region.

Table 9: Boat Navigation Experiment: Percentage Safe Episodes against additional baseline methods. Evaluated over 500 episodes and 5 seed values.

| Method | Safe Episodes (%) |
|---|---|
| BC+NCBF (Robey et al., 2020) | $88.43 \pm 0.59$ |
| CPQ (Xu et al., 2022) | $54.14 \pm 0.26$ |
| WSAC (Wei et al., 2024) | $79.34 \pm 0.87$ |
| CDT (Liu et al., 2023) | $69.94 \pm 0.42$ |
| C2IQL (LIU et al., 2025) | $70.77 \pm 0.76$ |
| BEAR-Lag (Kumar et al., 2019) | $71.69 \pm 0.16$ |
| COptiDICE (Lee et al., 2022) | $87.88 \pm 0.65$ |
| FISOR (Zheng et al., 2024) | $85.78 \pm 0.22$ |
| BC+**V-OCBF** (Ours) | $\mathbf{97.56 \pm 0.13}$ |

We compared V-OCBF against a broader set of baselines, including CPQ (Xu et al., 2022), WSAC (Wei et al., 2024), CDT (Liu et al., 2023), and C2IQL (LIU et al., 2025), in addition to the methods evaluated in the main text. The results, summarized in Table 9, demonstrate that V-OCBF consistently achieves a higher empirical safety rate compared to the baseline methods. Notably, the strongest baseline exhibits a safety rate approximately 9% lower than our approach. These results, combined with the experiments presented in Section 5, indicate that V-OCBF provides a robust mechanism for minimizing safety violations across diverse dynamical systems, effectively handling the challenges posed by nonlinear drift and learned dynamics.

# E  Experimental Details

## E.1  Experimental Hardware

To keep the evaluation fair and avoid any discriminatory added advantage to any specific experiment, all experiments were conducted on a single system equipped with an 14th Gen Intel Core i9-14900KS CPU, 128GB RAM, and an NVIDIA GeForce RTX 5090 GPU for training and experiment evaluations.

## E.2  Hyperparameters for the Proposed Algorithm

We have compiled and listed down all the hyperparameters that we used to perform our experiments and report the results. These training settings for all the environments are detailed in the Table 10. For the

Table 10: Hyperparameters for the Proposed (V-OCBF) Network, and Learned Dynamics Model.

| Hyperparameter | Value |
|---|---|
| Network Architecture | Multi-Layer Perceptron (MLP) |
| Activation Function | ReLU |
| Optimizer | Adam optimizer |
| Learning Rate | $3 \times 10^{-5}$ |
| Discount Factor ($\gamma$) | 0.99 |
| **Autonomous Ground Vehicle Collision Avoidance** | |
| Number of Hidden Layers | 2 |
| Hidden Layer Size | 256 neurons per layer |
| Dataset Size | 75K |
| Dynamics Model Hidden Layers | 3 |
| Dynamics Model Hidden Layer Size | 64 |
| **Safe Velocity Hopper** | |
| Number of Hidden Layers | 3 |
| Hidden Layer Size | 256 neurons per layer |
| Dataset Size | 1.32M |
| Dynamics Model Hidden Layers | 4 |
| Dynamics Model Hidden Layer Size | 64 |
| **Safe Velocity Half-Cheetah** | |
| Number of Hidden Layers | 3 |
| Hidden Layer Size | 128 neurons per layer |
| Dataset Size | 249K |
| Dynamics Model Hidden Layers | 4 |
| Dynamics Model Hidden Layer Size | 64 |
| **Safe Velocity Ant** | |
| Number of Hidden Layers | 3 |
| Hidden Layer Size | 256 neurons per layer |
| Dataset Size | 2.09M |
| Dynamics Model Hidden Layers | 4 |
| Dynamics Model Hidden Layer Size | 64 |
| **Safe Velocity Walker2D** | |
| Number of Hidden Layers | 3 |
| Hidden Layer Size | 256 neurons per layer |
| Dataset Size | 2.12M |
| Dynamics Model Hidden Layers | 4 |
| Dynamics Model Hidden Layer Size | 64 |
| **Safe Velocity Swimmer** | |
| Number of Hidden Layers | 3 |
| Hidden Layer Size | 256 neurons per layer |
| Dataset Size | 1.68M |
| Dynamics Model Hidden Layers | 4 |
| Dynamics Model Hidden Layer Size | 64 |

MuJoCo environments, we use the widely accepted DSRL Liu et al. (2024) dataset. We also use the *Class-K* function $\kappa = \alpha \times B(\cdot)$ where $\alpha = 1$.

### E.3  Hyperparameters for the Baselines

For all the CBF-based baselines (NCBF, iDBF, CCBF), we use the same network architectures and learned dynamics models described above to ensure a fair comparison. Hyperparameters for the safe offline RL baselines (COptiDICE, BEAR-Lag, BC, FISOR) are provided in Table 11. We use the official implementations of CPQ, CDT, BEAR-Lag and COptiDICE from (OSRL) Liu et al. (2024), WSAC from Wei et al. (2024),

C2IQL from LIU et al. (2025) and FISOR from Zheng et al. (2024), and train all methods on the same DSRL datasets.

Table 11: Hyperparameters for all the Baselines network. For COptiDICE and BEAR-Lag refer OSRL, Liu et al. (2024), for the official implementations. For FISOR refer Zheng et al. (2024) for official implementation.

| **Autonomous Ground Vehicle (AGV)** | |
|---|---|
| BC - Actor | MLPActor, (128,2) |
| BEAR-Lag - Actor | SquashedGaussianMLPActor, (128,2) |
| BEAR-Lag - Critic | EnsembleDoubleQCritic, (256,2) |
| BEAR-Lag - Cost critic | MLP, (256,2) |
| COptiDICE - Actor (extraction) | MLPActor, (128,2) |
| COptiDICE - Dual / $\nu$ network | DualNet (value-like), (256,2) |
| FISOR - Actor (diffusion denoiser backbone) | DiffusionDenoiserMLP, (128,2) |
| FISOR - Feasibility classifier | FeasibilityClassifier, (256,2) |
| **Safe Velocity Hopper** | |
| BC - Actor | MLPActor, (256,2) |
| BEAR-Lag - Actor | SquashedGaussianMLPActor, (256,2) |
| BEAR-Lag - Critic | EnsembleDoubleQCritic, (256,2) |
| BEAR-Lag - Cost critic | MLP, (256,2) |
| COptiDICE - Actor (extraction) | MLPActor, (256,2) |
| COptiDICE - Dual / $\nu$ network | DualNet (value-like), (256,2) |
| FISOR - Actor (diffusion denoiser backbone) | DiffusionDenoiserMLP, (256,2) |
| FISOR - Feasibility classifier | FeasibilityClassifier, (256,2) |
| **Safe Velocity Half-Cheetah** | |
| BC - Actor | MLPActor, (256,2) |
| BEAR-Lag - Actor | SquashedGaussianMLPActor, (256,2) |
| BEAR-Lag - Critic | EnsembleDoubleQCritic, (256,2) |
| BEAR-Lag - Cost critic | MLP, (256,2) |
| COptiDICE - Actor (extraction) | MLPActor, (256,2) |
| COptiDICE - Dual / $\nu$ network | DualNet (value-like), (256,2) |
| FISOR - Actor (diffusion denoiser backbone) | DiffusionDenoiserMLP, (256,2) |
| FISOR - Feasibility classifier | FeasibilityClassifier, (256,2) |
| **Safe Velocity Ant** | |
| BC - Actor | MLPActor, (256,2) |
| BEAR-Lag - Actor | SquashedGaussianMLPActor, (256,2) |
| BEAR-Lag - Critic | EnsembleDoubleQCritic, (256,2) |
| BEAR-Lag - Cost critic | MLP, (256,2) |
| COptiDICE - Actor (extraction) | MLPActor, (256,2) |
| COptiDICE - Dual / $\nu$ network | DualNet (value-like), (256,2) |
| FISOR - Actor (diffusion denoiser backbone) | DiffusionDenoiserMLP, (256,2) |
| FISOR - Feasibility classifier | FeasibilityClassifier, (256,2) |
| **Safe Velocity Walker2D** | |
| BC - Actor | MLPActor, (256,2) |
| BEAR-Lag - Actor | SquashedGaussianMLPActor, (256,2) |
| BEAR-Lag - Critic | EnsembleDoubleQCritic, (256,2) |
| BEAR-Lag - Cost critic | MLP, (256,2) |
| COptiDICE - Actor (extraction) | MLPActor, (256,2) |
| COptiDICE - Dual / $\nu$ network | DualNet (value-like), (256,2) |
| FISOR - Actor (diffusion denoiser backbone) | DiffusionDenoiserMLP, (256,2) |
| FISOR - Feasibility classifier | FeasibilityClassifier, (256,2) |
| **Safe Velocity Swimmer** | |
| BC - Actor | MLPActor, (256,2) |
| BEAR-Lag - Actor | SquashedGaussianMLPActor, (256,2) |
| BEAR-Lag - Critic | EnsembleDoubleQCritic, (256,2) |
| BEAR-Lag - Cost critic | MLP, (256,2) |
| COptiDICE - Actor (extraction) | MLPActor, (256,2) |
| COptiDICE - Dual / $\nu$ network | DualNet (value-like), (256,2) |
| FISOR - Actor (diffusion denoiser backbone) | DiffusionDenoiserMLP, (256,2) |
| FISOR - Feasibility classifier | FeasibilityClassifier, (256,2) |

