# OpenReview forum: "V-OCBF: Learning Safety Filters from Offline Data via Value-Guided Offline Control Barrier Functions"
_TMLR — Accepted by TMLR_

### Review · Reviewer_RsGZ · 2026-01-12

**Summary Of Contributions:**

This paper proposes Value-Guided Offline Control Barrier Functions (V-OCBF), a framework for synthesizing safety-certified controllers purely from offline demonstrations. The key idea is to learn a neural Control Barrier Value Function (CBVF) without access to system dynamics by introducing a finite-difference barrier recursion that propagates safety information across time using offline transitions. To avoid querying unsupported actions in the offline setting, the method incorporates an expectile-based objective that emphasizes high, data-supported safety values while remaining within the dataset action support. The learned barrier is then used in a standard CBF-QP formulation, where a separately learned dynamics model is employed only at inference time to compute Lie derivatives. Experiments on an autonomous ground vehicle task and high-dimensional Safety Gymnasium environments show that V-OCBF achieves fewer safety violations than constrained offline RL and prior offline neural CBF methods, while maintaining competitive reward.

**Strengths**

__1. Clear separation of learning and control__
The method clearly separates barrier learning from controller synthesis. The barrier is learned purely from offline data without using system dynamics, while learned dynamics are used only at inference time for QP-based safety filtering.

__2. Dynamics-free offline barrier learning__
The finite-difference barrier recursion enables learning a safety certificate without known dynamics or online interaction, making the approach suitable for offline and safety-critical settings.

__3. Strong empirical results across environments__
Experiments on both AGV and high-dimensional Safety Gymnasium tasks show consistently fewer safety violations than constrained offline RL and offline neural CBF baselines, with competitive reward.

__Weaknesses__

__1. Limited theoretical guarantees__
The formal guarantee is restricted to one-step forward invariance under the idealized barrier recursion, and long-horizon safety under function approximation is not theoretically established.

__2. Dependence on learned dynamics at inference__
Although dynamics are not used during barrier learning, the final safety filter relies on a learned dynamics model, and safety under model error is supported empirically rather than theoretically.

__3. Heuristic nature of expectile maximization__
Expectile regression is well-motivated but does not provide a formal guarantee of approximating the true maximization over admissible actions, and introduces an additional hyperparameter.

**Audience:**

Yes

**Audience Explanation:**

The paper addresses an important problem in safe decision making from offline data, which is relevant to researchers in offline reinforcement learning, control barrier functions, and safety-critical control. The focus on hard, state-wise safety guarantees and the separation between offline certificate learning and online control execution make the work particularly relevant to audiences interested in combining learning-based methods with control-theoretic safety mechanisms.

**Broader Impact Concerns:**

The work aims to improve safety in learning-based control systems, which is generally a positive contribution. No major ethical concerns are identified.

**Claims And Evidence:**

Yes

**Claims Explanation:**

The claims of the paper are supported by a combination of theoretical analysis and empirical evaluation. The finite-difference barrier recursion is formally derived from the CBVF formulation, and the paper proves that satisfying this recursion guarantees one-step forward invariance for control-affine systems. The motivation for expectile regression is clearly explained and aligns with prior work in offline reinforcement learning for avoiding out-of-distribution action evaluation. The experimental results span both low-dimensional and high-dimensional environments, include comparisons with strong constrained offline RL and offline neural CBF baselines, and are supported by ablation studies that validate key design choices. While the theoretical guarantees are limited in scope, they are consistent with what is claimed in the paper.

**Requested Changes:**

1. Clarify more explicitly the scope of the theoretical guarantees in the main text. In particular, emphasize that the formal result applies to one-step forward invariance under the idealized barrier recursion, and that closed-loop safety under function approximation and learned dynamics is not theoretically guaranteed.

2. Strengthen the positioning with respect to model-based offline RL. Since a dynamics model is learned and used at inference time, it would be helpful to clearly state that the learned dynamics are not used for policy optimization or imagined rollouts, but only for local Lie derivative computation in the CBF-QP.

3. Add a brief discussion in the related work section contrasting V-OCBF with model-based offline RL approaches, highlighting differences in objectives and the role of dynamics.

4. Provide a short intuitive explanation of how the expectile parameter affects conservatism in the learned barrier, to improve accessibility for readers less familiar with expectile regression.

---

> ### Author Response · Authors · 2026-02-01
>
> > **Scope of theoretical guarantees**
>
> We agree with the reviewer and have clarified the scope of our theoretical guarantees throughout the manuscript. Specifically, we now state that the formal result applies to one-step forward invariance under the idealized finite-difference barrier recursion, assuming exact barrier values and exact Lie-derivative computations. We also emphasize that closed-loop safety is not theoretically guaranteed in the presence of function approximation, finite-data effects, or learned-dynamics errors.
>
> These clarifications have been added in the Introduction, at the end of Section 4.3 (Controller Synthesis via Learned Dynamics), and in the Conclusion and Future Work section.
>
>
> > **Positioning w.r.t. model-based offline RL**
>
> We have strengthened the positioning of V-OCBF with respect to model-based offline RL in the Related Work / Positioning subsection (Section 2.3). The revised text now explicitly clarifies that, although a dynamics model is learned, it is *not* used for policy optimization, imagined rollouts, or barrier learning. Instead, dynamics are introduced only at inference time to compute local Lie derivatives required by the CBF-QP.
>
> We also added a brief contrast with model-based offline RL methods, highlighting that such approaches typically rely on learned dynamics during training to perform rollouts or planning, whereas V-OCBF propagates safety using a model-free value recursion and avoids querying unsupported state–action pairs. This distinction highlights both the different objectives and the inference-only role of dynamics in V-OCBF.
>
> > **Intuition for expectile parameter $\tau$**
>
> We have added an intuitive explanation of the role of the expectile parameter $\tau$ in Section 4.2 and Appendix A.2, motivated by insights from Implicit Q-Learning. The revised text explains that while prioritizing the best observed next-state outcomes can work well under ideal, noise-free conditions, directly focusing on the single best outcome is brittle in the presence of stochasticity or rare misleading transitions. We therefore clarify that expectile regression provides a practical alternative by softly emphasizing high-quality, dataset-supported outcomes rather than committing to extreme values, which improves robustness without querying unseen actions.
>
> To further aid understanding, we include Figure 5, which visualizes how varying $\tau$ affects the learned barrier and the resulting safe set. This figure helps illustrate how higher $\tau$ values reduce conservatism while remaining empirically well-behaved in our experiments.

---

### Review · Reviewer_FH4j · 2026-01-18

**Summary Of Contributions:**

The paper introduces Value-Guided Offline Control Barrier Functions (V-OCBF), a framework designed to synthesize formally safe controllers for autonomous systems using only static and offline datasets. Its primary contributions include model-free barrier synthesis, involving the derivation of a recursive finite-difference barrier update that allows learning a neural Control Barrier Function (CBF) without requiring prior knowledge of the system's analytical dynamics. Additionally, the framework implements an expectile-based learning objective using an expectile regression loss to approximate optimal safe-set expansion by focusing on the "upper envelope" of safe actions found in the data. This objective prevents the barrier from being distorted by out-of-distribution (OOD) actions while still improving upon the safety of the original behavior policy. The authors also propose decoupled dynamics learning, a training pipeline where a surrogate dynamics model is learned separately from the barrier function. These dynamics are utilized only during inference to compute Lie derivatives for a Quadratic Program (QP) safety filter to synthesize real-time safe control. Finally, the paper provides empirical validation demonstrating superior performance over existing safe offline RL methods and neural CBF baselines.


### Strengths
- Hard Safety Guarantees: Unlike many offline RL methods that treat safety as a "soft" penalty (minimizing expected cost), V-OCBF enforces hard, state-wise constraints through a QP formulation, providing formal one-step forward invariance.
- Practical Implementation: It functions as a "minimally invasive" safety filter, meaning it can be wrapped around any reference controller (like Behavior Cloning) to provide safety without requiring the reference controller itself to be safe.

### Weaknesses
- Inference-Time Model Reliance: While the barrier is learned model-free, the real-time safety filter (the QP) depends on a learned surrogate dynamics model. Significant inaccuracies in this learned model at inference time could lead to safety violations.
- Point-Wise Safety: The paper specifically proves one-step forward invariance, but managing safety over long horizons in highly stochastic environments may require further extensions.

**Audience:**

Yes

**Audience Explanation:**

TMLR's audience of researchers in safe reinforcement learning and control theory would find these findings highly relevant. The paper addresses a critical gap by providing hard, state-wise safety guarantees in offline settings where dynamics are unknown, which is a major challenge for autonomous systems like robotics and vehicles.

**Claims And Evidence:**

Yes

**Claims Explanation:**

The theoretical claim that the model-free finite-difference recursion preserves forward invariance is supported by a formal derivation and proof in the appendix. Convincing empirical evidence is provided through high-dimensional experiments in the MuJoCo Safety Gymnasium, where V-OCBF consistently achieves the lowest safety violation rates while maintaining competitive rewards compared to several baseline methods.

**Requested Changes:**

- Clarify Dependence on Learned Dynamics: The authors state that V-OCBF is "model-free" during barrier learning , yet a learned surrogate dynamics model is strictly required at inference to solve the Quadratic Program (QP). The authors must explicitly discuss the safety implications if the learned dynamics is not accurate enough.
 - V-OCBF vs. Model-Based Offline RL: Could you identify the difference between model-based offline RL and your proposed method? Also what is the pros and cons comparing between them?
- Figure 3 Analysis and Theory-Experiment Gap: Could you elaborate more on figure 3 about why there are more episode safety violations in Walker 2D compared with other environments in your proposed method? Also, since you claim that your method is hard safety constraints, then what is the gap between theory and experiment that there are still violations in practice?

---

> ### Author Response · Authors · 2026-02-01
>
> > **Dependence on Learned Dynamics at Inference Time**
>
> We agree with the reviewer’s concern and clarify that our claim of *model-free* applies to the learning stage and the synthesis of the safe value function. Extracting a safe policy from the learned value function requires solving a QP, which does involve an explicit dynamics model. Importantly, our empirical results suggest that a coarse or nominal dynamics model is often sufficient for this step in the evaluated environments. This assumption is practical, as approximate models are typically available in most robotic and autonomous systems.
>
> To empirically validate that approximate dynamics models suffice, we conduct an experiment on the Autonomous Ground Vehicle system to quantify the impact of dynamics model inaccuracies on safety performance. We consider three learned dynamics models with decreasing accuracy: (i) known dynamics, (ii) the model used in our main experiments, (iii) a model trained for half the number of training steps, and (iv) a model trained for one quarter of the training steps. We further evaluate model accuracy by computing mean prediction errors over 100 randomly sampled state–action pairs, confirming a monotonic degradation in predictive performance. Using each of these models, we then compute the resulting safety rate.
>
> **Table: Use of Learned Dynamics at the time of Inference (Mean over 500 episode rollouts).**
>
> | **Method** | **Safe Episodes (%)** | **Episode Reward** | **Mean L2 Model Error** |
> |------------|------------------------|--------------------|-------------------------|
> | BC+**V-OCBF** (Known Dynamics) | 99.52 ± 0.58 | 55.91 ± 0.39 | 0 |
> | BC+**V-OCBF** (Learned Dynamics) | 98.28 ± 0.54 | 54.93 ± 0.46 | $1.48 \times 10^{-2}$ |
> | BC+**V-OCBF** (Half-Learned Dynamics) | 97.03 ± 0.45 | 52.44 ± 0.18 | $2.93 \times 10^{-2}$ |
> | BC+**V-OCBF** (Quarter-Learned Dynamics) | 96.32 ± 0.73 | 51.91 ± 0.32 | $4.46 \times 10^{-2}$ |
>
> The results show that the safety rate degrades only marginally despite substantial reductions in dynamics accuracy, thereby suggesting that approximate dynamics models can be sufficient for safe policy extraction in the considered settings. We have added the above results in the Ablation Studies (Section 5.3 and Appendix C4).
>
> > **Comparison with Model-Based Offline Reinforcement Learning Approaches**
>
> Our method differs from model-based offline RL primarily in how and when the dynamics model is used. In V-OCBF, the control barrier function is learned using a model-free finite-difference recursion over offline transitions, without relying on a learned dynamics model during training. The dynamics model is introduced only at inference time, where it is used locally to compute Lie derivatives within the CBF-QP for safety filtering.
>
> In contrast, model-based offline RL methods typically rely on learned dynamics during training for imagined rollouts, value backups, or policy optimization. While this can improve sample efficiency, it also exposes learning to error accumulation and compounding model bias, which can degrade empirical safety performance in the offline setting. This effect is illustrated in Table 2 of the main paper, which reports the empirical safety rates obtained when learning a value-based CBF via the proposed model-free recursion compared to training the barrier using a learned dynamics model.
>
> The advantage of V-OCBF is that it avoids model error propagation during barrier learning, leading to more reliable empirical safety performance when learning from offline data. The trade-off is that a dynamics model is still required at deployment time, and safety under model mismatch is not theoretically guaranteed, though it is supported empirically in our experiments.
>
> > **Empirical Safety Violations and the Theory–Practice Gap**
>
> The increased number of safety violations in Walker2D can be attributed to the higher dimensionality and more complex contact dynamics of the environment, which make accurate barrier approximation more challenging under function approximation. Since the CBF is learned using deep neural networks from offline data, approximation errors and limited coverage of critical transition regions can lead to occasional constraint violations in practice.
>
> Regarding the gap between theory and experiments, we have revised the manuscript to clarify that the formal safety guarantees hold only under idealized conditions, assuming exact satisfaction of the barrier recursion and perfect model information. In practical implementations, learning errors, finite data, and approximation inaccuracies introduce deviations from these assumptions, which can result in residual violations.
>
> We have updated the conclusion and future work sections to explicitly state that providing stronger guarantees, using statistical methods (like Conformal Prediction) or Lipschitz-based robustness guarantees for learned CBFs, remains an important direction for future research.

---

### Review · Reviewer_4BQn · 2026-01-23

**Summary Of Contributions:**

**Summary**: This paper considers learning a safety filter from offline data for a value-guided control barrier function. Traditional methods use the known dynamics, or grid search, to find the safety filter; in this paper, they use offline data and the value function to learn the action. In particular, they use the offline data to learn the V-OCBF, and also learns a dynamic model, so that during the inference time, a safe action can be taken. The paper then provides empirical validation.

**Strength**:

1. The problem is timely, as using offline data would be of great importance, and the approach is practical as it is grounded in previous works.
2. The approach has shown improvement over the others.

**Weakness**:

1. The main weakness is the lack of technical novelty. The paper itself has pointed out that learning the value-based control barrier function is inspired by the Fisac et al.  There are plenty of approaches for learning the control barrier function [A1--A4]. The only contribution is to use the offline data. However, the techniques proposed in this paper are not new, they have already been proposed. The adaptation is new, though for learning the control barrier function. However, it is not clear whether that would be a valid technical contribution.

2. The paper uses a safety filter-based approach; however, there are plenty of papers on using a safe offline RL-based approach. The paper has not compared with those works. The reviewer is aware of the fact that they are two different tools, but they are solving the same problem. For example, one can put the cost in the unsafe region to be 1, and in the safe region to be 0. One can pose this problem as a constrained MDP problem. There are existing offline approaches [A5--A9] to solve this problem effectively.

3. Even though the empirical results show improvement, the improvement is mild over the existing ones. Can the authors identify one experiment where the existing approaches' results are poor, and the approach proposed by the authors show a significant improvement?

[A1]. Robey, Alexander, Haimin Hu, Lars Lindemann, Hanwen Zhang, Dimos V. Dimarogonas, Stephen Tu, and Nikolai Matni. "Learning control barrier functions from expert demonstrations." In 2020 59th IEEE Conference on Decision and Control (CDC), pp. 3717-3724. Ieee, 2020.

[A2]. Taylor, Andrew, Andrew Singletary, Yisong Yue, and Aaron Ames. "Learning for safety-critical control with control barrier functions." In Learning for dynamics and control, pp. 708-717. PMLR, 2020.

[A3]. Emam, Yousef, Gennaro Notomista, Paul Glotfelter, Zsolt Kira, and Magnus Egerstedt. "Safe reinforcement learning using robust control barrier functions." IEEE Robotics and Automation Letters (2022).

[A4]. Wabersich, Kim P., and Melanie N. Zeilinger. "Predictive control barrier functions: Enhanced safety mechanisms for learning-based control." IEEE Transactions on Automatic Control 68, no. 5 (2022): 2638-2651.

[A5]. Xu, H., Zhan, X., and Zhu, X. Constraints penalized q-learning for safe offline reinforcement learning. Proceedings of the AAAI Conference on Artificial Intelligence, 36(8):8753–8760, Jun. 2022. doi: 10.1609/aaai.v36i8.20855.

[A6]. Wei, H., Peng, X., Ghosh, A., and Liu, X. Adversarially trained weighted actor-critic for safe offline reinforcement
learning. Advances in Neural Information Processing Systems, 37:52806–52835, 2024.

[A7]. Liu, Z., Guo, Z., Yao, Y., Cen, Z., Yu, W., Zhang, T., and Zhao, D. Constrained decision transformer for offline safe reinforcement learning. In International Conference on Machine Learning, pp. 21611–21630.

[A8]. Wang, Y., Zhan, S.S., Jiao, R., Wang, Z., Jin, W., Yang, Z., Wang, Z., Huang, C. and Zhu, Q., 2023, July. Enforcing hard constraints with soft barriers: Safe reinforcement learning in unknown stochastic environments. In International Conference on Machine Learning (pp. 36593-36604). PMLR.

[A9]. Zifan, L. I. U., Xinran Li, and Jun Zhang. "C2IQL: Constraint-Conditioned Implicit Q-learning for Safe Offline Reinforcement Learning." In Forty-second International Conference on Machine Learning.

**Audience:**

Yes

**Audience Explanation:**

The paper is useful for the robotics, and the control community.

**Claims And Evidence:**

No

**Claims Explanation:**

Even though using an offline RL technique to learn the control barrier function might be new, the tools are not new, which limits the contributions. Further, the approach has not compared with the safe offline RL approaches. Hence, the paper needs a significant revision.

**Requested Changes:**

1. Can you please identify the main technical novelties and difficulties in extending the existing offline approaches to learn the value-based control barrier function?

2. Can the authors compare with the existing safe offline RL approaches, as the reviewer pointed out?

3. Can the authors identify one experiment where the existing approaches' results are poor, and the approach proposed by the authors show a significant improvement?

---

> ### Author Response · Authors · 2026-02-01
>
> > **Technical Novelty and Challenges in Learning Value-Based Control Barrier Functions Offline**
>
> A key challenge in extending existing offline methods to value-based control barrier functions lies in the fundamentally myopic nature of classical CBF formulations, which enforce safety through one-step forward invariance rather than long-horizon safety propagation. Most existing approaches are therefore not directly compatible with value-based or recursive formulations.
>
> Early learning-based CBF methods such as [A1] rely on known system dynamics to supervise the barrier learning process, which limits their applicability to systems with unknown or partially known dynamics. In contrast, our approach learns a value-based CBF purely from offline transition data without requiring access to dynamics during training.
>
> Furthermore, several prominent works [A2,A3,A4] do not learn CBFs from data, but instead assume analytically specified barrier functions, which are difficult to design and do not scale to complex or high-dimensional systems with unknown dynamics. Finally, recent approaches such as [A8] explicitly restrict their applicability to non-hybrid systems, whereas many practical safety-critical systems exhibit hybrid or contact-rich dynamics.
>
> Our main technical novelty is addressing these limitations by introducing a value-based, dynamics-free barrier learning formulation that propagates safety information through offline data, enabling scalable CBF learning from offline data under unknown dynamics.
>
> > **Comparison with Existing Safe Offline Reinforcement Learning Methods**
>
> Existing safe offline RL approaches differ from our method primarily in how safety is incorporated during learning and deployment. Many offline safe RL methods encode safety through penalty terms or constrained objectives [A5,A9], where the degree of safety enforcement depends on careful tuning of penalty coefficients and is evaluated empirically rather than through explicit safety filtering mechanisms based on learned barrier functions.
>
> Learning-based CBF approaches provide a complementary perspective, but existing methods either assume access to known system dynamics during training [A1] or rely on analytically specified barrier functions [A2,A3,A4], which can be difficult to design and scale to complex systems with unknown dynamics. Additionally, some approaches explicitly restrict applicability to non-hybrid systems [A8] or rely on online interaction, which may be impractical in safety-critical offline settings.
>
> Our approach differs by learning a control barrier value function directly from offline transition data, without requiring known dynamics during training. A learned dynamics model is used only at inference time to support safety filtering via a CBF-QP. While this does not provide formal safety guarantees, our empirical results show improved safety performance across multiple environments compared to existing safe offline RL and offline CBF-based baselines. We further provide a comparative analysis with representative safe offline RL methods (including [A1,A5,A6,A7,A9]) in the subsequent response and accompanying table, which we kindly ask the reviewer to consider.
>
> > **Additional Experimental Evidence Highlighting Performance Gains**
>
> While our method consistently achieves the highest safety rates across the benchmark environments, we agree that highlighting a concrete failure mode of existing approaches strengthens the empirical evaluation. To this end, we include an additional boat navigation task with nonlinear river drift and obstacle constraints, which is challenging for existing safe offline RL and CBF-based baselines.
>
> This environment is low-dimensional and interpretable, yet exhibits nonlinear dynamics that significantly affect safety, making it particularly difficult for methods relying on soft constraints or learned dynamics during training. Full environment details and dynamics are provided in Appendix D of the revised manuscript.
>
> The resulting safety performance is summarized below (evaluated over 500 episodes and 5 random seeds):
> | Method | Safe Episodes (%) |
> |--------|-------------------|
> | BC+NCBF [A1] | 88.43 ± 0.59 |
> | CPQ [A5] | 54.14 ± 0.26 |
> | WSAC [A6] | 79.34 ± 0.87 |
> | CDT [A7] | 69.94 ± 0.42 |
> | C2IQL [A9] | 70.77 ± 0.76 |
> | BEAR-Lag [A10] | 71.69 ± 0.16 |
> | COptiDICE [A11] | 87.88 ± 0.65 |
> | FISOR [A12] | 85.78 ± 0.22 |
> | **BC+V-OCBF (Ours)** | **97.56 ± 0.13** |
>
> Across all baselines, our method achieves a substantially higher empirical safety rate, with at least a 9\% absolute improvement over the strongest competing method. This experiment provides a clear example where existing approaches exhibit degraded safety performance, while our method maintains significantly improved empirical safety. The full experimental setup and additional analyses are included in Section 5.3 and Appendix D.

---

> > ### Author Response · Authors · 2026-02-01
> > **References**
> >
> > ## References
> >
> > [A1]. Robey, Alexander, Haimin Hu, Lars Lindemann, Hanwen Zhang, Dimos V. Dimarogonas, Stephen Tu, and Nikolai Matni. "Learning control barrier functions from expert demonstrations." In 2020 59th IEEE Conference on Decision and Control (CDC), pp. 3717-3724. Ieee, 2020.
> >
> > [A2]. Taylor, Andrew, Andrew Singletary, Yisong Yue, and Aaron Ames. "Learning for safety-critical control with control barrier functions." In Learning for dynamics and control, pp. 708-717. PMLR, 2020.
> >
> > [A3]. Emam, Yousef, Gennaro Notomista, Paul Glotfelter, Zsolt Kira, and Magnus Egerstedt. "Safe reinforcement learning using robust control barrier functions." IEEE Robotics and Automation Letters (2022).
> >
> > [A4]. Wabersich, Kim P., and Melanie N. Zeilinger. "Predictive control barrier functions: Enhanced safety mechanisms for learning-based control." IEEE Transactions on Automatic Control 68, no. 5 (2022): 2638-2651.
> >
> > [A5]. Xu, H., Zhan, X., and Zhu, X. Constraints penalized q-learning for safe offline reinforcement learning. Proceedings of the AAAI Conference on Artificial Intelligence, 36(8):8753–8760, Jun. 2022. doi: 10.1609/aaai.v36i8.20855.
> >
> > [A6]. Wei, H., Peng, X., Ghosh, A., and Liu, X. Adversarially trained weighted actor-critic for safe offline reinforcement learning. Advances in Neural Information Processing Systems, 37:52806–52835, 2024.
> >
> > [A7]. Liu, Z., Guo, Z., Yao, Y., Cen, Z., Yu, W., Zhang, T., and Zhao, D. Constrained decision transformer for offline safe reinforcement learning. In International Conference on Machine Learning, pp. 21611–21630.
> >
> > [A8]. Wang, Y., Zhan, S.S., Jiao, R., Wang, Z., Jin, W., Yang, Z., Wang, Z., Huang, C. and Zhu, Q., 2023, July. Enforcing hard constraints with soft barriers: Safe reinforcement learning in unknown stochastic environments. In International Conference on Machine Learning (pp. 36593-36604). PMLR.
> >
> > [A9]. Zifan, L. I. U., Xinran Li, and Jun Zhang. "C2IQL: Constraint-Conditioned Implicit Q-learning for Safe Offline Reinforcement Learning." In Forty-second International Conference on Machine Learning.
> >
> > [A10] Aviral Kumar, Justin Fu, Matthew Soh, George Tucker, and Sergey Levine. Stabilizing off-policy q-learning
> > via bootstrapping error reduction. Advances in neural information processing systems, 32, 2019.
> >
> > [A11] Jongmin Lee, Cosmin Paduraru, Daniel J Mankowitz, Nicolas Heess, Doina Precup, Kee-Eung Kim, and
> > Arthur Guez. COptiDICE: Offline constrained reinforcement learning via stationary distribution correction
> > estimation. In International Conference on Learning Representations, 2022.
> >
> > [A12] Yinan Zheng, Jianxiong Li, Dongjie Yu, Yujie Yang, Shengbo Eben Li, Xianyuan Zhan, and Jingjing Liu. "Safe offline reinforcement learning with feasibility-guided diffusion model." In International Conference on Learning Representations, 2024.

---

> > > ### Comment · Reviewer_4BQn · 2026-02-02
> > > **Concerns remain**
> > >
> > > I thank the authors for their efforts. However, my concerns have not been fully resolved. Thus, unfortunately, I am recommending rejection. I also feel that this paper is better suited for L4DC or Control journals rather than ML research, as the main contribution seems to be using offline RL tools for solving control barrier functions. But, that is not the main reason for this recommendation.
> > >
> > > 1. The authors mentioned that their main technical contribution is developing `` value-based, dynamics-free barrier learning formulation that propagates safety information through offline data, enabling scalable CBF learning from offline data under unknown dynamics."
> > >
> > > However, that contribution is not novel. In particular, (8) has been proposed before in the reachability literature. Please see [A1-A3]. I agree that they have not been specific in using offline data. However, how to use the offline data has been extensively studied, and the paper has not contributed to this aspect. Thus, the contributions are limited with respect to the TMLR.
> > >
> > > 2. Regarding the offline safe RL,
> > >
> > > It is true that constrained MDPs indeed consider a soft constraint (as they consider a cumulative constraint). However, one can denote unsafe states as $1$, and safe states as $0$, then impose the constraint as $V_c^{\pi}\leq 0$ to ensure that the policy is always safe. Hence, I think that offline constrained RL contains this problem. Further, the paper did not talk about the hyperparameter tuning for the other offline safe RL algorithms.
> > >
> > > [A1]. Fisac, J.F., Lugovoy, N.F., Rubies-Royo, V., Ghosh, S. and Tomlin, C.J., 2019, May. Bridging hamilton-jacobi safety analysis and reinforcement learning. In 2019 International Conference on Robotics and Automation (ICRA) (pp. 8550-8556). IEEE.
> > >
> > > [A2]. Hsu, K.C., Rubies-Royo, V., Tomlin, C.J. and Fisac, J.F., 2021. Safety and liveness guarantees through reach-avoid reinforcement learning. arXiv preprint arXiv:2112.12288.
> > >
> > > [A3]. Yu, D., Ma, H., Li, S. and Chen, J., 2022, June. Reachability constrained reinforcement learning. In International conference on machine learning (pp. 25636-25655). PMLR.

---

> > > > ### Author Response · Authors · 2026-02-02
> > > > **Response to the Remaining Concerns**
> > > >
> > > > > **On Value-based Safety Propagation and Reachability**
> > > >
> > > > We agree with the reviewer that recursive, value-based safety formulations have strong connections to reachability analysis and prior online reinforcement learning–based reach-avoid methods [A1–A3]. Our work does not aim to introduce a new reachability theory, nor do we claim conceptual primacy over these approaches. Instead, our contribution lies in adapting value-based safety propagation to a strictly offline setting and integrating it with control barrier function–based safety filtering at inference time. In contrast to prior reachability and reach-avoid RL methods, which typically rely on online interaction, V-OCBF learns safety values purely from offline transition data, avoids imagined rollouts or PDE solvers during training, and enables deployment via a CBF-QP that enforces state-wise constraints during execution.
> > > >
> > > > This setting is particularly relevant for safety-critical applications, where direct online interaction with the environment is often infeasible due to safety risks, and where the persistent sim-to-real gap continues to undermine the reliability of policies trained purely in simulation. These challenges motivate the need for offline learning frameworks that can synthesize safe control policies in a model-free manner while explicitly reasoning about worst-case safety violations over the entire time horizon, rather than relying on soft safety constraints. To the best of our knowledge, the closest prior work addressing this problem is [A1] from the reviewer's main review (“Learning Control Barrier Functions from Expert Demonstrations”). While [A1] represents an important step toward offline safety learning, our approach overcomes several of its limitations, as discussed in our previous response, and consistently achieves stronger empirical performance across the evaluated settings.
> > > >
> > > > > **Offline constrained RL and safety through cumulative violations**
> > > >
> > > > In safety-critical control, safety is inherently governed by the worst-case behavior over time rather than by cumulative or average violations. This perspective is standard in the reachability and Hamilton--Jacobi safety literature [A1,A2,A3], where safety values are defined through extremal (min / infimum) operators that capture whether a single unsafe state is ever encountered along a trajectory.
> > > >
> > > > Most offline constrained RL methods, in contrast, formulate safety via cumulative violation costs, which represent an expected or discounted aggregation of violations. While expressive, this formulation fundamentally differs from worst-case state-wise safety and does not directly yield a geometric safe set suitable for local action filtering. This distinction is reflected empirically in our results, where cumulative-violation-based offline RL methods exhibit substantially lower state-wise safety rates compared to our extremal, value-based CBF formulation.
> > > > Additionally, to further support our position, we find it helpful to directly quote reference [A3], which was cited by the reviewer and whose observation is consistent with the viewpoints expressed in [A1] and [A2]. Specifically, [A3] notes that “Safety problems in reality are more about the worst-case through time other than the cumulative or average costs, where the latter is often the case in previous CRL,” highlighting a key limitation of many prior CRL formulations. This perspective closely aligns with our argument and reflects a widely recognized consideration in safety-critical systems, namely that CRL methods based on cumulative safety violation costs can lead to weaker safety performance when worst-case behavior over time is the primary concern.
> > > >
> > > > > **Hyperparameter Tuning of Baseline Offline SafeRL Algorithms**
> > > >
> > > > We assure the reviewer that the baselines were not run with arbitrary hyperparameters. We utilized the OSRL (Offline Safe Reinforcement Learning) benchmark library, which provides tuned, community-standard hyperparameters specifically optimized for the Safety Gymnasium tasks used in our evaluation. We utilized these official configurations to ensure a fair and rigorous comparison. We have added this clarification to the experimental setup in the revised manuscript.

---

> > > > > ### Author Response · Authors · 2026-02-02
> > > > > **Scope and Positioning of the Paper**
> > > > >
> > > > > We acknowledge the reviewer's suggestion regarding control venues. However, we respectfully argue that TMLR is the appropriate venue because our primary objective is solving a fundamental *machine learning* problem: learning safe control policies from offline datasets. In our framework, Control Barrier Functions (CBFs) serve merely as a tool, an intermediate representation, to achieve this goal, rather than being the final theoretical object of study. Unlike standard offline constrained RL methods that optimize for "soft" expected costs (often failing to prevent specific unsafe occurrences), we utilize this barrier-based structure to enforce stricter, state-wise safety constraints within the learning process. This contribution is primarily of interest to researchers in **Safe Offline RL and Embodied AI**, who require methods that aim for worst-case constraint satisfaction. As noted in the reviewer's initial assessment (which identified the paper as relevant to the TMLR audience), this work bridges the gap between these fields, making TMLR the natural home for this research.
> > > > >
> > > > > We thank the reviewer again for their thoughtful comments, which have helped us sharpen the paper's scope, positioning, and clarity. We would be happy to further clarify any remaining questions or concerns the reviewer may have and sincerely appreciate their reconsideration of the rating or verdict should these clarifications resolve the issues raised.

---

> > > > > ### Comment · Reviewer_4BQn · 2026-02-13
> > > > > **Still learning for rejection**
> > > > >
> > > > > I thank the authors for their latest response. I still don't understand why the CMDP framework cannot contain a chance constraint where the cost is only state-dependent. For other types of constraints, I can understand the difference. Let me explain, suppose I want that $\Pr(\pi \text{ is safe})\geq 1-\delta$, then we can define the cost function as $c(s)=0$ when the state is safe, and $c(s)=1$ when the state is unsafe. Then, the constraint is typically $\mathbb{E}_{\pi}[ \mathbb{1}(\sum_tc_t\geq 0)]\leq \delta$. So, you can augment the state with this indicator function and transform it into a CMDP problem. I can understand that using a barrier function can help you in effectively solving the problem, but it is unclear when the dynamics are not known, which one would be a good choice.

---

> > > > > > ### Author Response · Authors · 2026-02-13
> > > > > >
> > > > > > We appreciate the reviewer’s engagement and the specific suggestion to use an indicator-based objective $E_{\pi}[\mathbb{1}(\sum c_t\ge0)]\le\delta$. However, we respectfully argue that this formulation faces fundamental theoretical and practical challenges in safe control synthesis that our approach specifically aims to resolve:
> > > > > >
> > > > > > - Continuous Signal vs. Binary Classification: The indicator-based objective collapses the rich state space information into a binary outcome, making it unable to distinguish between degrees of safety. For example, consider a trajectory that merely grazes the boundary versus a trajectory that passes directly through the middle of the unsafe region. The indicator function would mark both as equally unsafe (value = 1). This deprives the policy of the necessary information to distinguish "bad" from "worse." In contrast, our Lipschitz continuous safety function $\ell(x)$ assigns a significantly lower (more negative) value to the trajectory deep in the unsafe region compared to the one grazing the boundary. This provides the policy with a stronger, informative signal to push the agent away from the core of the unsafe region, rather than treating all failures as identical. This is consistent with the safety-critical control literature, where safety constraints are typically modeled using Lipschitz-continuous signed distance functions, while indicator functions are generally avoided due to their lack of regularity.
> > > > > >
> > > > > > - Worst-Case Safety and Gradient Compatibility: In safety-critical situations, ensuring robust safety requires valid gradients to steer the system, which the indicator function fails to provide. The indicator-based value function approximates a binary step function (0 for safe, 1 for unsafe), resulting in vanishing gradients almost everywhere except at the immediate boundary, where the gradient becomes a sharp, discontinuous jump tending toward infinity. This lack of informative gradients prevents the QP (or policy gradients) from generating any preemptive control signal, causing the safety filter to be myopic and failing to act until a violation is imminent or already occurring. By using a Lipschitz continuous safety barrier function $\ell(x)$, as also advocated in the works [A1-A3] cited by the reviewer, we ensure non-zero gradients that actively guide the agent toward safety well before the boundary is reached.
> > > > > >
> > > > > > Finally, regarding the reviewer's concern about unknown dynamics, we have explicitly demonstrated in our paper that our method works efficiently on systems with unknown dynamics, including the AGV, Boat Navigation, and high-dimensional Safety Gymnasium environments. In these experiments, we do not assume knowledge of the dynamics beforehand, further validating the applicability of our approach to the very settings the reviewer is concerned about.
> > > > > >
> > > > > > Furthermore, if the reviewer has specific prior works in mind that successfully implement the suggested indicator-based approach for continuous control synthesis without encountering the gradient issues discussed above, we would appreciate the references.

---

### Review · Reviewer_P4mu · 2026-02-16

**Summary Of Contributions:**

The paper proposes V-OCBF, a framework to learn a neural Control Barrier Function (CBF) / Control-Barrier Value Function (CBVF) from offline demonstrations without assuming access to true system dynamics. The method derives a model-free finite-difference recursion for the barrier (a reachability-style Bellman backup) and trains a neural barrier using a discounted regression objective to avoid trivial fixed-point solutions. To mitigate the offline “out-of-distribution action” issue, the paper adopts an expectile regression objective (inspired by IQL) to bias the learned barrier toward high (safe) values supported by the dataset’s action support, without querying barrier targets on unseen actions. At deployment, the learned barrier is used in a CBF-QP safety filter; since Lie derivatives are needed, a separate learned control-affine dynamics model is trained from the dataset and used only at inference to compute the QP terms.

**Audience:**

Yes

**Audience Explanation:**

Learned CBF is an important topic in safe control.

**Broader Impact Concerns:**

None.

**Claims And Evidence:**

Yes

**Claims Explanation:**

- Clear problem motivation: the paper targets strict, state-wise safety (forward invariance) rather than expected-cost constraints, and focuses on offline settings where online exploration is infeasible.
- Methodological clarity: the finite-difference barrier recursion and the two-stage learning procedure (behavior-induced backup then value-guided/expectile shaping) are explained cleanly, with algorithms and a workflow diagram.
- Addresses a key offline pitfall: the expectile objective is a reasonable, practical mechanism to avoid maximizing over actions not supported by the dataset, while still pushing beyond a purely behavior-induced barrier.
- Empirical results are fairly extensive for the offline-safety setting: evaluations include a low-dimensional collision-avoidance task (AGV Dubins car) and multiple high-dimensional Safety Gymnasium MuJoCo tasks. Reported metrics include safety violations, reward, and a “safe set volume” proxy.
- The paper includes ablations: effects of using learned dynamics in training vs inference, robustness to barrier network size, robustness to perturbations/noise, and sensitivity to expectile level tau.

**Requested Changes:**

1) Detailed questions for the authors

Q1. Guarantee statement: What precise property is claimed when the learned B_theta only approximately satisfies the finite-difference recursion (due to function approximation and sampling)? Do you measure recursion residuals, and do they correlate with violations?

Q2. Action-support construction: How do you implement U_D(x) in continuous spaces? Is it kNN over states, or do you use all actions in a minibatch? Please provide exact details and complexity.

Q3. QP feasibility: When the QP is infeasible (due to control limits or model error), what is the fallback? Do you soften constraints with slack variables, and if so, how is safety assessed?

Q4. Learned dynamics: How large are the dynamics model errors (one-step prediction error) in each environment, and is there a relationship to safety violations? Do you use ensembles or uncertainty estimation?

Q5. Expectile tau: The ablation suggests tau near 1 gives best safety in AGV. Does very large tau ever lead to overly conservative behavior or degraded reward on MuJoCo tasks? A per-task tau sensitivity plot would be useful.

2) Suggestions for improvement
- Clarify and narrow the formal claim: explicitly state whether the guarantee is discrete-time one-step invariance, and under what assumptions it extends to multi-step invariance.
- Add a short section on deployment-time robustness to model error (even if only empirical): e.g., dynamics ensembles, uncertainty-aware margins in the QP constraint, or a discussion of robust CBF-QP variants.
- Provide concrete implementation details for approximating dataset action support U_D(x) and a small sensitivity check (k in kNN, bandwidth, etc.).
- Extend evaluation with at least one additional constraint type beyond velocity thresholds, or a clear discussion of how the approach scales to multi-constraint / geometry-based safety.

---

> ### Author Response · Authors · 2026-02-20
>
> > Guarantee Statement
>
> We thank the reviewer for pointing this out. The formal one-step forward invariance guarantee holds only under idealized conditions, specifically, assuming exact satisfaction of the finite-difference barrier recursion and perfect system model information. We acknowledge that in practical implementations, closed-loop safety is not theoretically guaranteed due to function approximation, finite data effects, and learned-dynamics errors. We do not explicitly measure recursion residuals and correlate them with violations in the current scope of this work. However, we agree that approximation errors and limited coverage of critical transition regions by the neural network are the primary drivers of the residual constraint violations observed in practice. We have updated the Introduction and Section 4.3 to explicitly state these limitations regarding the scope of our theoretical guarantees.
>
> > Action Support Construction
>
> We thank the reviewer for requesting clarification on this important implementation detail. To answer, we do not use k-Nearest Neighbors (kNN) over states to explicitly construct the action support. Instead, drawing inspiration from Implicit Q-Learning (IQL), we implement the expectile regression directly over the empirical state-action transitions present in the sampled minibatch. By evaluating the expectile loss strictly over the $(x,u,x')$ tuples within a training minibatch, the algorithm naturally enforces the dataset support constraint.
>
> Therefore, as it operates directly on the minibatch transitions, the computational complexity scales linearly with the batch size, making it highly efficient. To ensure this is clear to readers, we have revised Algorithm 1 to explicitly show the expectile loss evaluated over the minibatch $\mathcal{B}$, and we have added explanatory text in Section 4.2 mentioning this practical implementation.
>
> > Clarification on One-Step invariance and Discussion on Multi-Step invariance
>
> We thank the reviewer for their comment. Our formulation guarantees discrete-time one-step invariance under the idealized assumption of zero learning or approximation error. To obtain multi-step invariance guarantees, one would additionally need to assume execution of the optimal safe policy obtained by maximizing the associated Hamiltonian:
> \begin{equation}
> H(x) = \max_{u \in \mathcal{U}} \langle \nabla B_{\theta}(x), f(x) + g(x) u \rangle.
> \end{equation}
> This leads to the corresponding optimal safe control law:
> \begin{equation}
> \pi_{\theta}(x) = \arg \max_{u \in \mathcal{U}} \langle \nabla B_{\theta}(x), f(x) + g(x) \rangle.
> \end{equation}
>
> However, as evident from the expressions above, computing this optimal safe policy requires exact knowledge of the system dynamics, which is not available in our setting. The resulting policy is highly sensitive to the sign of $(\nabla B\cdot g(x))$, implying that even small errors in the learned or approximate dynamics can lead to incorrect control selection and potential safety violations.
> Consequently, this Hamiltonian-based formulation lacks robustness to modeling and learning errors in the dynamics. For this reason, we adopt a quadratic program (QP)-based formulation, which still preserves discrete-time one-step invariance guarantees while offering improved robustness under approximate dynamics. We have updated the Introduction and Section 4.3 to explicitly state these limitations regarding the scope of our theoretical guarantees.
>
> > Discussion on QP Feasibility
>
> We thank the reviewer for raising the concern regarding QP feasibility. Formally, the safety guarantees of the standard QP-based CBF formulation hold in the absence of control input constraints, i.e., under the assumption of unbounded actuation. In practice, when control constraints are present, the CBF literature addresses potential infeasibility through the introduction of slack variables, which relax the safety constraint while penalizing violations. We adopt this standard formulation in our experiments. Importantly, we observe that the slack variables in our experiments remain close to zero. This is because the learned safe value function explicitly accounts for control constraints. Consequently, the learned policy exhibits a high level of safety compliance while remaining within the admissible control bounds, which significantly reduces the frequency with which the QP constraint becomes active. Even when the QP is triggered, the corrective adjustment is typically small, leading to minimal slack usage that remains within admissible control limits.
>
> To clarify this aspect, we have explicitly included the slack-variable formulation in Appendix A.3 of the revised manuscript.

---

> > ### Author Response · Authors · 2026-02-20
> >
> > > Impact of Learned Dynamics
> >
> > We agree with the reviewer’s observation regarding the potential impact of dynamics model inaccuracies on safety. In our experiments, we observe that the learned safe policy remains reasonably robust to moderate errors in the learned dynamics. To empirically validate this, we conduct an experiment on the Autonomous Ground Vehicle system to quantify the impact of dynamics model inaccuracies on safety rate. We consider three learned dynamics models with decreasing accuracy: (i) known dynamics, (ii) the model used in our main experiments, (iii) a model trained for half the number of training steps, and (iv) a model trained for one quarter of the training steps. We further evaluate model accuracy by computing mean prediction errors over 100 randomly sampled state–action pairs, confirming a monotonic degradation in predictive performance. Using each of these models, we then compute the resulting safety rate.
> >
> > **Table:** Use of Learned Dynamics at the time of Inference (Mean over 500 episode rollouts).
> >
> > | **Method** | **Safe Episodes (%)** | **Episode Reward** | **Mean L2 Model Error** |
> > | :--- | :--- | :--- | :--- |
> > | BC+**V-OCBF** (Known Dynamics) | 99.52 $\pm$ 0.58 | 55.91 $\pm$ 0.39 | 0 |
> > | BC+**V-OCBF** (Learned Dynamics) | 98.28 $\pm$ 0.54 | 54.93 $\pm$ 0.46 | $1.48\times10^{-2}$ |
> > | BC+**V-OCBF** (Half-Learned Dynamics) | 97.03 $\pm$ 0.45 | 52.44 $\pm$ 0.18 | $2.93\times10^{-2}$ |
> > | BC+**V-OCBF** (Quarter-Learned Dynamics) | 96.32 $\pm$ 0.73 | 51.91 $\pm$ 0.32 | $4.46\times10^{-2}$ |
> >
> > The results show that the safety rate degrades only marginally despite substantial reductions in dynamics accuracy, thereby suggesting that approximate dynamics models can be sufficient for safe policy extraction in the considered settings. We have added the above results in the Ablation Studies (Section 5.3 and Appendix C4). Finally, we note that our current implementation does not employ ensemble-based dynamics models or explicit uncertainty quantification. Incorporating uncertainty-aware techniques, such as conformal prediction for quantifying model error, represents an interesting and promising direction for future work.
> >
> > > Additional Experiments to clarify the role of $\tau$ Expectile
> >
> > We thank the reviewer for raising the important question regarding the effect of the expectile parameter $\tau$ on safety and performance. Under idealized conditions (deterministic dynamics and no dataset disturbances, i.e., when the best actions reliably lead to the best next states), pushing the expectile $\tau \rightarrow 1$ can yield strong empirical performance, prioritizing high-value next-state targets. However, in environments with stochastic disturbances, setting $\tau$ very close to $1$ may result in suboptimal behavior. Due to stochasticity, suboptimal actions in the dataset can occasionally transition to high-value states, and a near-extreme expectile may overemphasize such transitions. This can degrade both robustness and safety. To mitigate this effect, $\tau$ must be chosen sufficiently high to prioritize strong targets, yet not so close to $1$ that it becomes overly sensitive to stochastic noise in the dataset.
> >
> > To empirically validate this reasoning, we conducted an additional set of experiments evaluating the impact of $\tau$ across multiple Safety MuJoCo tasks using standard, public datasets from DSRL, which are not strictly deterministic. We evaluated both safety (measured via Cost) and task performance (measured via Reward) for $\tau \in \{0.5, 0.7, 0.9, 0.99\}$, as shown in Table below.
> >
> > **Table:** Reward and Cost across different $\tau$ expectile values for selected MuJoCo environments
> >
> > | Environment | $\tau=0.5$ Reward | $\tau=0.5$ Cost | $\tau=0.7$ Reward | $\tau=0.7$ Cost | $\tau=0.9$ Reward | $\tau=0.9$ Cost | $\tau=0.99$ Reward | $\tau=0.99$ Cost |
> > | :--- | :--- | :--- | :--- | :--- | :--- | :--- | :--- | :--- |
> > | Ant | 565.36 | 0.0 | 639.61 | 0.0 | 791.77 | 0.0 | 734.66 | 0.02 |
> > | Half Cheetah | 100.39 | 0.0 | 180.87 | 0.0 | 679.73 | 0.0 | 582.58 | 0.05 |
> > | Hopper | 96.94 | 0.0 | 126.54 | 0.0 | 769.69 | 0.0 | 526.72 | 0.27 |
> >
> > As observed in Table above, increasing $\tau$ toward $1$ (specifically $\tau = 0.99$) results in degraded safety performance, reflected by increased Cost, and in some cases reduced Reward. This trend is consistent across the reported tasks and aligns with our theoretical intuition regarding sensitivity to stochastic transitions. In contrast, $\tau = 0.9$ achieves a more favorable balance between reward maximization and safety preservation. Similar patterns were observed across other environments. Based on these findings, we selected $\tau = 0.9$ for the experiments reported in the main paper. We have incorporated this additional analysis into the revised manuscript (Appendix C.3).

---

> > > ### Author Response · Authors · 2026-02-20
> > >
> > > > Evaluation on multi-constraint /geometry-based safety task
> > >
> > > We agree that evaluating performance in multi-constraint environments provides a stronger empirical assessment of safety-critical behavior. To this end, we have extended our experimental evaluation to include an additional boat navigation task. Specifically, we consider a 2D autonomous boat navigating a river with state-dependent drift, where the objective is to reach a target island while avoiding two circular obstacles of different radii. These obstacles induce multiple geometric safety constraints, and the nonlinear river drift introduces state-dependent dynamics that can actively push the agent toward unsafe regions if not properly accounted for. This environment is particularly challenging for methods that enforce safety through soft constraints, as cumulative constraint penalties may not adequately prevent instantaneous violations under adverse drift dynamics. The results are illustrated in the Table below:
> > >
> > > | **Method** | **Safe Episodes (%)** |
> > > | :--- | :--- |
> > > | BC+NCBF | 88.43 $\pm$ 0.59 |
> > > | CPQ | 54.14 $\pm$ 0.26 |
> > > | WSAC | 79.34 $\pm$ 0.87 |
> > > | CDT | 69.94 $\pm$ 0.42 |
> > > | C2IQL | 70.77 $\pm$ 0.76 |
> > > | BEAR-Lag | 71.69 $\pm$ 0.16 |
> > > | COptiDICE | 87.88 $\pm$ 0.65 |
> > > | FISOR | 85.78 $\pm$ 0.22 |
> > > | BC+**V-OCBF** (Ours) | **97.56** $\pm$ **0.13** |
> > >
> > > Across all baselines, our method achieves a substantially higher empirical safety rate (97.56\%), with at least a 9\% absolute improvement over the strongest competing baseline. These results demonstrate the effectiveness of our approach in handling multi-constraint, geometry-driven safety tasks under nonlinear dynamics. The full experimental setup, system dynamics, and results for this multi-constraint environment have been added to Section 5.3 and Appendix D.

---

### Author Response · Authors · 2026-02-01

We sincerely thank the reviewers for their insightful and constructive reviews, which significantly improved the clarity and quality of the paper. We have responded to all points below and marked the relevant changes in blue in the revised manuscript.

---

### Decision · Action_Editor_QznM · 2026-03-10

**Recommendation:** Accept with minor revision

**Additional Comments:**

The paper already states that the learned barrier is not a formally verified certificate and that approximation and learned-dynamics errors can break guarantees in practice. The authors should make that caveat much more prominent in the abstract, intro, contributions, and conclusion, instead of writing “formally safe from offline demonstrations” in their contribution item 1.

**Audience:**

Yes

**Audience Explanation:**

Yes, this paper would be of interest to researchers working on safe reinforcement learning and learning for control.

**Claims And Evidence:**

Yes

**Claims Explanation:**

This paper proposes a framework that learns a neural control barrier function purely from offline data, without needing the system dynamics during training. The learned barrier function is then used as a safety filter that modifies any controller at runtime.
The main advantages are that: no online interaction required and it handles the case of unknown dynamics. The proposed method is similar to IQL and other papers that make use of an expectile regression objective in how it handles OOD states.

Overall, reviewers' initial concerns have been addressed with additional experiments and clarifications and they recommend acceptance of the paper.

---

> ### Author Response · Authors · 2026-04-04
>
> We sincerely thank the Action Editor and the reviewers, who have helped us refine the paper. In response, we have revised the manuscript to make the limitations of V-OCBF more explicit and to ensure that no statement suggests formal guarantees beyond what is supported in practice.